# Comparative single-cell transcriptomic analysis of primate brains highlights human-specific regulatory evolution

Hamsini Suresh [1], Megan Crow[2], Nikolas Jorstad[3], Rebecca Hodge[3], Ed Lein[3], Alexander Dobin [1], Trygve Bakken [3] & Jesse Gillis [1,4] ✉

Enhanced cognitive function in humans is hypothesized to result from cortical expansion and increased cellular diversity. However, the mechanisms that drive these phenotypic innovations remain poorly understood, in part because of the lack of high-quality cellular resolution data in human and non-human primates. Here, we take advantage of single-cell expression data from the middle temporal gyrus of five primates (human, chimp, gorilla, macaque and marmoset) to identify 57 homologous cell types and generate cell type-specific gene co-expression networks for comparative analysis. Although orthologue expression patterns are generally well conserved, we find 24% of genes with extensive differences between human and non-human primates (3,383 out of 14,131), which are also associated with multiple brain disorders. To assess the functional significance of gene expression differences in an evolutionary context, we evaluate changes in network connectivity across meta-analytic co-expression networks from 19 animals. We find that a subset of these genes has deeply conserved co-expression across all non-human animals, and strongly divergent co-expression relationships in humans (139 out of 3,383, <1% of primate orthologues). Genes with human-specific cellular expression and co-expression profiles (such as *NHEJ1*, *GTF2H2*, *C2* and *BBS5*) typically evolve under relaxed selective constraints and may drive rapid evolutionary change in brain function.

Cortical expansion and increased cellular diversity in the human brain following divergence from great apes are hypothesized to contribute to enhanced cognitive function[1,2], but the molecular mechanisms underlying human brain evolution are not fully understood. High protein sequence conservation between humans and non-human primates suggests that cortical evolution in the human lineage is driven primarily by changes in the regulation of gene expression[3,4]. Comparative cross-species transcriptomic analyses are essential to uncover gene expression programmes underlying cell identity[5–7], and assess the impact of their dysregulation in neuropsychiatric disease[8,9]. Difficulty in obtaining and preserving samples, and the quality of genome annotation in non-human primates have restricted the scope of most comparative studies in primates to characterizing patterns of gene regulation across a small set of species using bulk transcriptomic data from a limited number of tissues[10–14]. Moreover, recent analyses[15,16] highlight the difficulty of disentangling functional gene co-regulation

¹Stanley Institute for Cognitive Genomics, Cold Spring Harbor Laboratory, Cold Spring Harbor, NY, USA. ²Genentech, South San Francisco, CA, USA. ³Allen Institute for Brain Science, Seattle, WA, USA. ⁴Department of Physiology, University of Toronto, Toronto, Ontario, Canada. ✉e-mail: jesse.gillis@utoronto.ca

confounded with co-expression because of variation in cell-type abundance across tissue samples. Comparative co-expression analysis at single-cell resolution has the potential to systematically trace the origin and diversity of cell types across animal evolution.

Single-cell transcriptomics has become a powerful tool to identify regional and interspecific variation in gene expression underlying the evolution of brain regions and cell types within[17–19] and across species[20–24]. For example, aligning human and mouse samples from homologous brain regions revealed extensive divergence in gene expression of cortical cell types[2], and the presence of a primate-specific striatal interneuron population[25], highlighting the need to study primate brains at high resolution to uncover the mechanisms behind evolutionary innovations in the human lineage. To identify human-specific patterns of gene activity driving brain evolution, we used the gene expression resource recently generated by the BRAIN Initiative Cell Census Network consortium[26]. This dataset contains high-quality single-nucleus transcriptomic atlases of the middle temporal gyrus (MTG) sampled from five primates (human, chimp, gorilla, macaque and marmoset), spanning an evolutionary period of ~45 million years. The essence of our approach was to identify cell types shared across species, and then use this common sample space to determine where and how orthologues change their expression pattern.

In this study, we identified 57 homologous cell types by aligning single-nucleus MTG atlases of five primates. We observed high cross-species similarity in expression variability over 57 consensus cell types for orthologous genes, suggesting conserved transcriptional patterning across primates. However, we also noted that a substantial fraction of genes had divergent expression variation between human and non-human primates in one or more cell classes. These genes were enriched for synapse assembly and function, and nearly half showed expression divergence limited to glial cell types. Because changes in gene expression can evolve under random drift or natural selection, we assessed the functional impact of expression variation by investigating changes in network connectivity using gene co-expression networks spanning a phylogenetically diverse set of 19 animals from CoCoCoNet[27]. Despite tissue heterogeneity in individual RNA sequencing (RNA-seq) datasets, we previously demonstrated that we can produce reliable estimates of gene co-expression through large-scale meta-analysis of publicly available gene expression data[28]. Using high-powered co-expression networks, we also established that genes typically have highly conserved co-expression neighbourhoods across evolutionarily distant species, highlighting their conserved regulation and function[28]. We identified 139 genes with divergent gene expression and connectivity exclusive to the human lineage, which also displayed a higher tolerance to inactivation, suggesting evolution under relaxed mutational constraint as a key driver of human-specific gene activity.

Overall, we generate a comparative catalogue of gene expression data over 57 matched cell types from MTG of five primates, and make this available through a web-based resource (https://gillislab.shinyapps.io/Primate_MTG_coexp/) for further exploration. We also demonstrate that integrative analysis of gene expression at single-cell resolution with cross-species co-expression conservation (represented schematically in Fig. 1) is a powerful approach to distinguish evolutionarily conserved transcriptional features from uniquely human gene expression traits.

## Results

### Consensus MTG taxonomy across primates

The BRAIN Initiative Cell Census Network[26] generated high-resolution transcriptomic maps of the MTG in human, chimpanzee, gorilla, macaque and marmoset by applying single-nucleus transcriptomic (snRNA-seq) assays to samples isolated from between three and seven donor brains in each species (plate-based SMART-seq v4 (SSv4) for great apes, in addition to droplet-based Chromium v3 (Cv3) RNA-seq for all primates). Because gene expression signatures at the cell subclass level

are highly reproducible across species and brain regions[23], primate MTG datasets were annotated by transferring subclass labels from the human primary motor cortex taxonomy[23].

In total, 574,156 nuclei passed quality control, including 341,469 excitatory (glutamatergic) neurons, 158,188 inhibitory (GABAergic) neurons and 74,499 non-neuronal cells (Fig. 2a). Cells in each species were categorized into three classes (non-neurons, excitatory and inhibitory neurons) and 24 subclasses. Datasets were integrated across individuals and data modalities (SSv4, Cv3) in each species, and the integrated space was subdivided into cell-type clusters using a previously described 'shatter and merge' approach[23] (further details are given in the Methods section 'snRNA-seq processing and clustering'). This approach resulted in a varied number of cell-type clusters across species, ranging from 103 in marmoset to 151 in humans. Next, we assessed the replicability of cell types at different levels of granularity across species using MetaNeighbor[29,30], which identifies cell types with highly similar transcriptional signatures within and across species. Cell types at the class and subclass levels of annotation were near-perfectly replicable across species, confirming that cell types have distinct transcriptomic profiles that distinguish them at broad levels of cell classification (Fig. 2b,e; refer to the Methods section 'Replicability of clusters' for additional details on measuring cell-type reproducibility). However, at finer resolution, multiple clusters exhibited substantial transcriptomic similarities within and across species (Supplementary Table 1). Because we are interested in assessing the conservation of gene expression signatures across matched cell types between human and non-human primates, we first generated a comprehensive set of homologous cell types (cross-species clusters) as described below.

First, we applied MetaNeighbor to identify highly replicable clusters across species, which formed the initial pool of consensus cell types. Next, we used a weighted nearest-neighbour approach to assign each of the remaining ambiguously matched clusters to the consensus cell type containing the majority of transcriptionally similar cell clusters (Fig. 2c; see the Methods section 'Replicability of clusters' for more details). This clustering procedure allowed us to map 594 clusters in all five primates to 86 cross-species clusters, with each cross-species cluster containing one or more clusters from at least two primates. All primates shared 57 of 86 cross-species clusters (Supplementary Fig. 1). We refer to these shared clusters as homologous cell types, and they contain more than 80% of clusters from each species (Fig. 2d). As expected, homologous cell types showed similar transcriptional profiles across species (Fig. 2f).

To assess the reliability of cross-species cluster assignment, we permuted through all possible combinations of leave-one-gene-out cross-validation, and predicted sets of best-matched cell types across all folds for each pair of species. Clustering results were generally consistent with consensus cell types defined by the initial clustering pipeline (mean adjusted Rand index = 0.999), confirming the robustness of the generated taxonomy. We also functionally characterized our consensus clusters by identifying HUGO Gene Nomenclature Committee (HGNC)- and Synaptic Gene Ontology (SynGO)-curated gene groups that contributed the most to replicability[30]. Genes related to cell adhesion and neuronal signalling were most informative of cell-type identity, and showed similar classification performance when trained and tested in the same or different species (Fig. 2g; scores for all 920 gene groups are listed in Supplementary Table 2).

### Characterizing gene expression patterns across primates

We organized the 57 homologous cell types into a hierarchical taxonomy on the basis of transcriptomic similarities (Fig. 3a), and observed that hierarchical relationships among cell types roughly mirrored their developmental origins. This consensus taxonomy provides an excellent opportunity to infer the extent of functional conservation between humans and non-human primates by comparing the similarity of gene expression signatures across matched cell types.

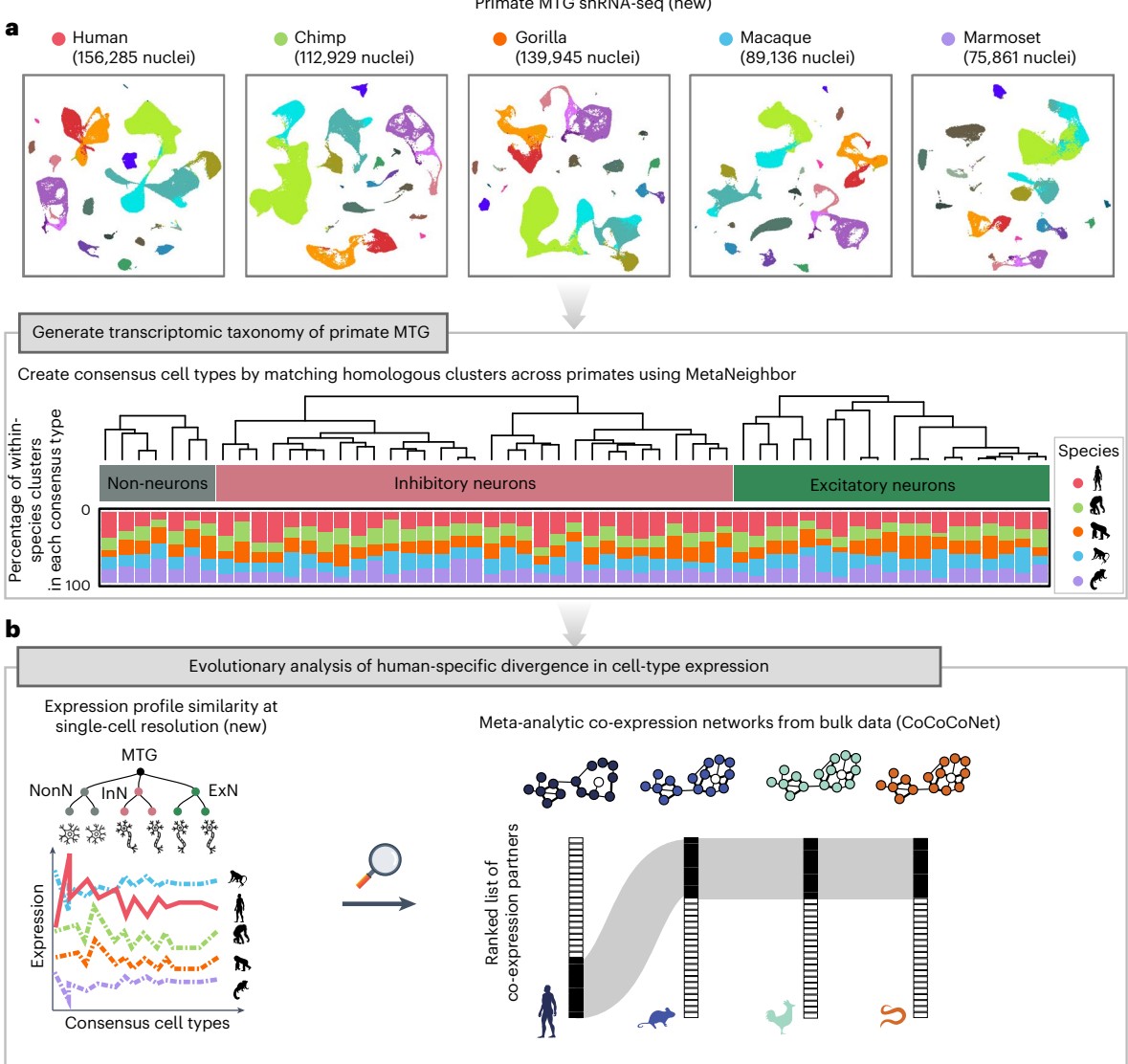

**Fig. 1 | Schematic illustration of evolutionary analysis to infer subtle regulatory shifts underlying human-specific single-cell transcriptomic divergence. a**, We used the single-nucleus transcriptomic atlases of the MTG of five primates to create a consensus classification of MTG cell types. The bar plot shows the percentage of within-species cell-type clusters associated with each consensus cell type, coloured by species. **b**, We quantified the similarity of gene expression profiles across primates to select genes with conserved expression signatures across non-human primates but diverged in humans (left). We then tested for signatures of differential regulation driving human-specific cell-type expression profiles by measuring changes in co-expression network connectivity between humans and 18 animals sampled broadly across metazoan phylogeny (right). The figure indicates that the gene retains its top ten co-expression partners in all animals except humans, suggesting that differential co-expression connectivity could underlie human-specific expression divergence. Silhouettes for all five primates are from www.phylopic.org (public domain). ExN, excitatory neuron; InN, inhibitory neuron; NonN, non-neuron.

Adopting the language of Patel et al.[31], given a query gene from one species, the homologous gene in the target species with the most similar expression variability across a set of matched tissues is referred to as its 'expressolog'. Although this method has been employed to select functionally similar orthologues from homologous gene clusters, we apply it here to evaluate the similarity of expression profiles of one-to-one (1:1) orthologues compared with that of random gene pairs. We obtained a list of 14,131 human genes with 1:1 orthologues in all non-human primates from OrthoDB[32]. For each pair of species, we calculated the expression profile similarity for all pairs of genes by correlating the mean normalized expression levels across 57 homologous cell types. We then defined the rank-standardized expression profile similarity of 1:1 orthologues relative to all other genes as the 'expressolog score' (Fig. 3b; see the Methods section 'Calculating the expressolog score' for details on the calculation). In essence, this measures whether orthologues show similar expression profiles across cells. This score is represented as an area under the receiver operating characteristic curve (AUROC) with a score of 1 signifying specific and highly similar expression variation across the species pair, 0.5 indicating dissimilar/uncorrelated expression variation and 0 indicating significant extreme expression profile divergence in one or both species. Our earlier clustering cross-validation ensured that no gene can drive the cluster assignments to ensure a high expressolog score for itself.

The expressolog score for each gene measures the specificity with which transcriptional signatures across shared cell types can be used to detect its 1:1 orthologues across species. Intuitively, an expressolog score of 0.99 for a gene indicates that its orthologue is in the top 1% of all genes in terms of expression profile similarity. Because genes

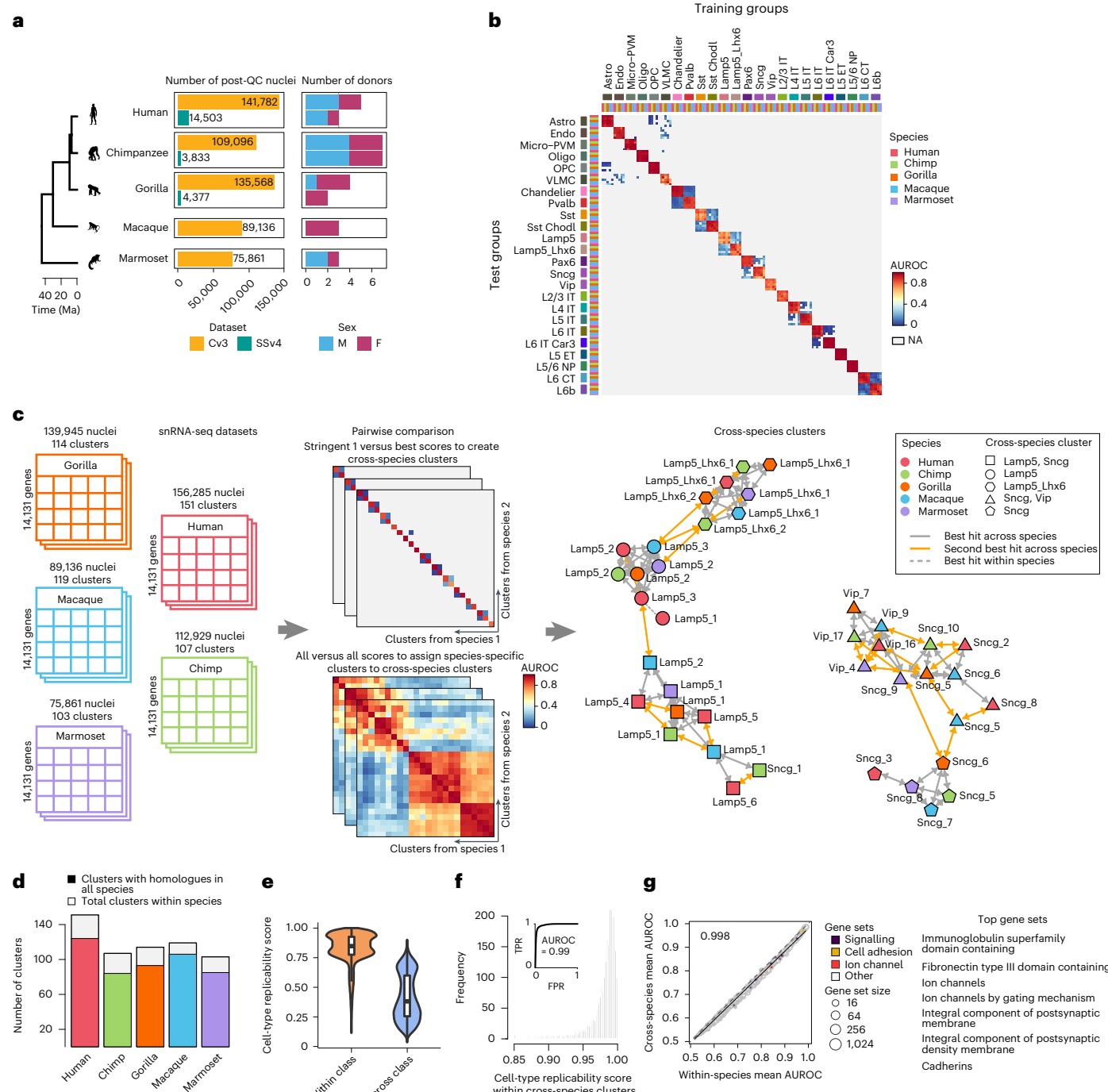

**Fig. 2 | Homologous cell types across five primates. a**, Summary of single-nucleus transcriptomic data split by sequencing technology, number and sex of donors for each species. Ma, million years ago. **b**, Heatmap showing the 'one_vs_best' MetaNeighbor scores for cell subclasses across primates, with cell types labelled by species and subclass. Each column shows the performance of a single training group across the five test datasets. Cell subclass replicability scores (AUROCs) were computed between the two closest neighbours in each test dataset, where the closer neighbour has the higher score (shown in red; all others are shown in grey). NA, not available. **c**, Scheme showing a semisupervised MetaNeighbor framework used to define consensus transcriptomic cell types across primates. **d**, Fraction of cell types from each species in the consensus MTG taxonomy. **e,f**, Cross-species clustering of cell types is validated by comparing

cell-type reproducibility within and across cell classes (*n* = 594 clusters) (**e**) and plotting the distribution of cell-type replicability scores for matched clusters across species (**f**). The receiver operating characteristic curve in the inset indicates cluster replicability score identifies consensus cell types with high fidelity. **g**, Scatter plot depicts the performance of 920 HGNC- and SynGO-curated gene groups in classifying consensus cell types within and across species, coloured by functional category. Linear regression fit is indicated by the black line, with the slope in the upper left-hand corner. Top highly conserved gene sets across primates are listed on the right (cell-type classification performance >0.95). For all boxplots, the bounds of the box represent the first and third quartiles, the thick line represents the median and the whiskers represent 1.5× the interquartile range.

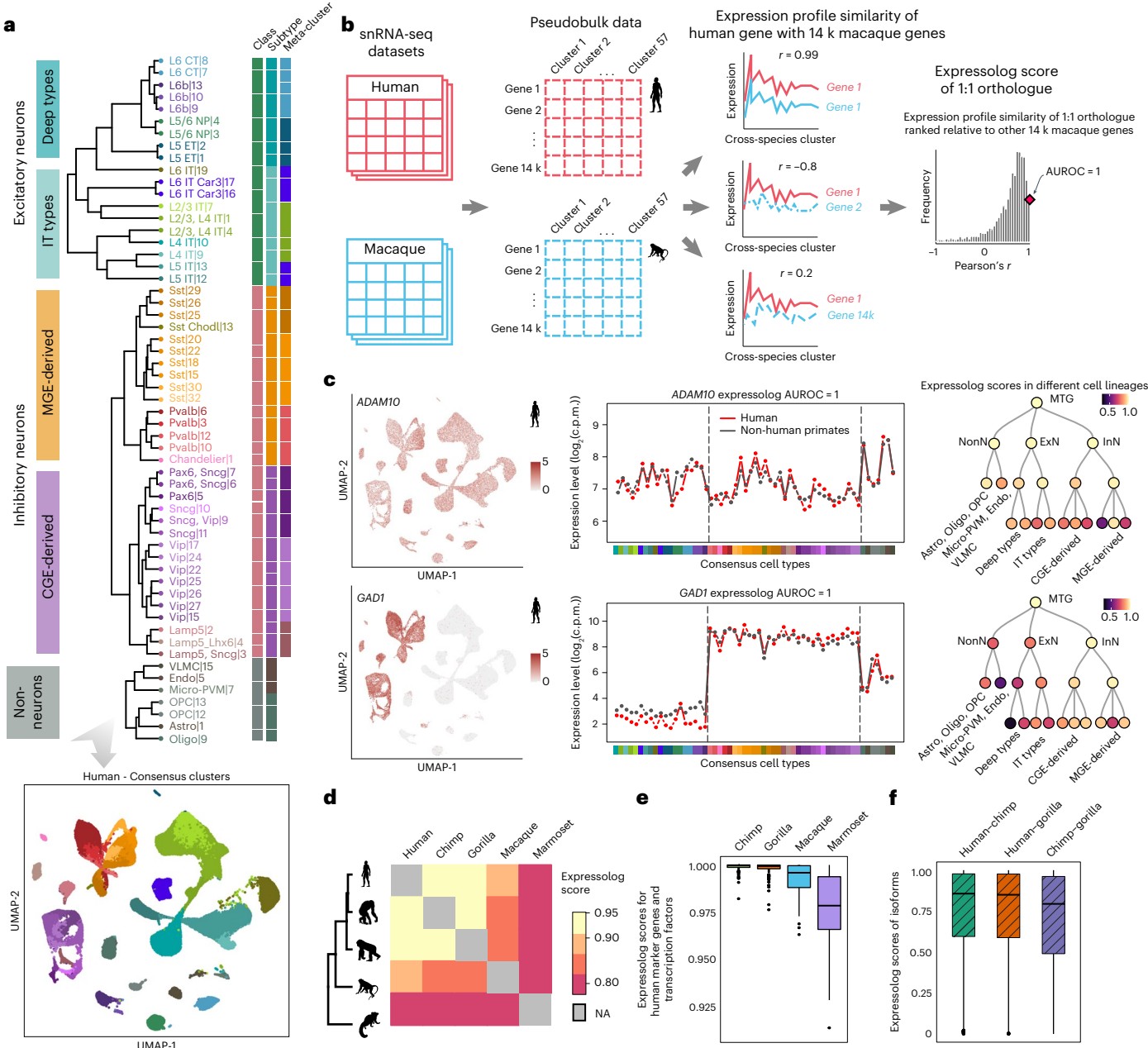

**Fig. 3 | Orthologues have conserved expression profiles across primates and the extent of conservation recapitulates known phylogeny. a**, Dendrogram of 57 consensus cell types defined by their transcriptomic similarity, annotated with corresponding cell class, subtype and meta-cluster (upper), and UMAP plot of single nuclei from human MTG integrated across donors and snRNA-seq technologies and coloured by consensus cluster (lower). CGE, caudal ganglionic eminence; IT, intralencephalic; MGE, medial ganglionic eminence. **b**, Method used to calculate the expression profile similarity of 1:1 orthologues for a pair of species. **c**, Examples of genes with constitutive (*ADAM10*) and cell type-specific expression (*GAD1*) in the human MTG data (colour indicates expression level in the UMAP plots). Both genes have near-identical expression profiles between human and non-human primates (expressolog score = 1 in both cases). Expressolog scores computed across cell types within each subgroup reveal persistent transcriptomic similarities at different levels of granularity (with

the exception of *Pvalb* cell subclass for *ADAM10*, and L5/6 excitatory neurons and vascular cells for *GAD1*). Astro, astrocyte; c.p.m., counts per million; Endo, endothelial cell; Oligo, oligodendrocyte; OPC, oligodendrocyte precursor; PVM, perivascular macrophage; VLMC, vascular and leptomeningeal cell. **d**, Heatmap of expressolog scores averaged across 14,131 orthologues for all pairs of primates. **e**, Boxplots indicate that cell lineage-specific genes like marker genes distinguishing the three cell classes and transcription factors have conserved expression profiles across primates suggesting conserved transcriptional programmes shape cell identity across species (*n* = 128 genes). **f**, Expressolog scores suggest that individual isoforms also exhibit similar expression profiles across great apes (*n* = 8,190 isoforms). For all boxplots, the bounds of the box represent the first and third quartiles, the thick line represents the median and the whiskers represent 1.5× the interquartile range.

with shared functions often display similar expression profiles, we used expressolog scores computed over 57 matched cell types as a measure of gene functional conservation across species. We found that orthologous genes show highly similar patterns of expression variation

across cell types and are highly conserved across the phylogeny. Two such examples are shown in Fig. 3c: *ADAM10*, which is constitutively expressed in the primate MTG, and *GAD1*, which is expressed exclusively in inhibitory neurons. Both genes exhibit perfectly matched expression

profiles across human and non-human primates, corresponding to an expressolog score of 1 in each case. Notably, both *ADAM10* and *GAD1* display conserved patterns of expression variation even across more homogeneous cell types (such as caudal ganglionic eminence (CGE)-derived interneurons or deep layer excitatory neurons), suggesting that the genes are deeply conserved across species.

Early microarray-based comparative studies noted that the divergence in gene activity in the same tissue between species reflects their evolutionary relationships[12,33]. Similarly, average expressolog score (or cross-species co-expression) correlates with evolutionary distances between human and non-human primates (Fig. 3d; refer to Supplementary Table 3 for the full list of scores). Expressolog scores correctly classify orthologues with performance ranging from 0.93 for humans with great apes, to 0.8 for humans with marmoset. Marker genes and transcription factors also show high functional conservation across species, suggesting a highly conserved molecular landscape of the MTG region across primates (Fig. 3e; see Methods for details on marker and transcription factor selection).

Alternative splicing is known to increase transcriptomic diversity in primates[34,35], but the functional conservation of individual isoforms is yet to be fully characterized. Do isoforms have reproducible transcriptional signatures across primates? To address this, we used SSv4 data with full transcript coverage in 28 cell types in human, chimp and gorilla to explore patterns of isoform usage across great apes. In general, isoforms showed similar expression profiles across species (Fig. 3f). For each gene with multiple isoforms in a pair of species, we calculated the expressolog scores for all isoform pairs to measure the ability of each isoform to correctly predict itself across species. Overall performance for this task was slightly better than that expected by chance (AUROC = 0.56), suggesting similar but not specific transcriptional patterning of isoforms across species. Indeed, consistent with previous observations[35], we also noted extensive isoform switching across apes, which could explain the weak expression specificity of isoforms across species.

## Assessing gene activity conservation using co-expression

Does transcriptional similarity across primates correspond to conserved gene regulation? Because gene co-expression reflects shared regulation and function, we can assess the conservation of regulatory mechanisms underlying shared transcriptional patterning by quantifying the similarity of gene co-expression neighbourhoods across species[28,36–38]. We built gene co-expression networks for each of the five primates by aggregating individual cell type-specific co-expression networks built from pseudobulk samples of consensus cell types (see left-hand panel of Fig. 4a for a schematic representation, and refer to the Methods section 'Building aggregate co-expression networks' for further details on building co-expression networks).

To measure the similarity of gene co-expression neighbourhoods between human and non-human primates, we subset gene co-expression networks to 4,500 highly variable 1:1 orthologues, and calculated a 'co-expression conservation' score, which is a measure of gene neighbourhood replicability across the species pair (see right-hand panel of Fig. 4a for a schematic representation, and the Methods section 'Calculating cross-species co-expression conservation' for further details). We observed that gene co-expression neighbourhoods are highly conserved across primates (Fig. 4b), revealing a highly conserved cellular architecture of the MTG region in primates. Similar to expression profile similarity, we found that co-expression neighbourhood similarity also correlates with primate phylogeny.

The reliability of gene functional conservation estimation is limited by the statistical power of the underlying co-expression networks. Currently, we need greater single-cell sequencing depth for better transcriptomic coverage, and multiple high-quality datasets that can be aggregated to build well-powered cell type-specific co-expression networks to link changes in gene expression to cell type-specific regulatory rewiring. Therefore, co-expression network analysis using large-scale aggregation of bulk expression data remains important for studying evolutionary differences driving species-specific transcriptional signatures. However, we need to evaluate the conservation of gene co-expression across (1) different levels of cell-type granularity, and (2) divergent species before we can leverage the vast amounts of publicly available bulk RNA-seq data to pinpoint species-specific regulatory changes that could be associated with divergent expression patterning in the MTG.

Are gene co-expression relationships replicable across networks built at different levels of cell-type heterogeneity? We can now compare co-expression networks from snRNA-seq datasets with networks derived from whole-brain or cross-tissue samples in humans to distinguish co-expression due to shared co-regulation from co-expression driven by cell-type composition. We generated a meta-analytic human brain co-expression network by aggregating datasets of human bulk brain data sourced from the Gemma database[39] (Fig. 4c). At coarser resolution, we also obtained a high-confidence human gene co-expression network from CoCoCoNet[27] created by meta-analysis of publicly available bulk RNA-seq datasets. Co-expression neighbourhood conservation calculated pairwise between the three aggregate co-expression networks revealed functional conservation of genes at different levels of cell-type resolution (Fig. 4c). The high degree of consistency between single-nucleus and bulk networks highlights conserved co-regulatory relationships across tissues and cell types. This observation is consistent with a model of multiscale co-expression in the brain, which proposes that cell types may differ substantially in gene expression levels, but share a core co-regulatory network.

Are gene co-expression relationships conserved across metazoa? Although the neocortex is a feature specific to mammals, the basic regulatory components underlying functional changes there may have evolved before mammalian evolution and undergone extensive re-organization in different phylogenetic classes[40,41]. Therefore, we can use co-expression conservation of functionally relevant genes to test for signs of conserved molecular identity across species. Ideally, we would like to test this idea through meta-analysis of large-scale brain-specific transcriptomic data from multiple species, but such data are available only for select model species. Previously, we showed that our gold standard human gene co-expression network (assembled from datasets sampling multiple bulk tissues) is topologically and functionally similar to our meta-analytic brain-specific human network (Fig. 4c), capturing the key regulatory features shared by both. Based on this observation, we tested the conservation of neuronal and non-neuronal cell-type marker genes using bulk co-expression networks of humans and 21 other species available on CoCoCoNet (networks derived from 54,668 samples over 22 species as reported in Supplementary Table 5). We observe consistently high co-expression conservation scores of 1:1 orthologues even in phylogenetically distant species like fruit fly and roundworm (Fig. 4d), suggesting the presence of conserved regulatory features across metazoa. We note that this result is robust to different marker gene selection criteria (Supplementary Fig. 2) and is also recapitulated using brain-specific co-expression networks (Supplementary Fig. 3), highlighting the replicability of gene–gene relationships across diverse species and tissues.

We observe that the two complementary measurements of gene functional conservation−expression profile similarity between human and non-human primates and co-expression conservation between human and 21 other species−are broadly consistent with each other (Fig. 4e). We also note that genes with divergent expression profiles between human and non-human primates exhibit greater changes in co-expression network connectivity in humans, indicating that an integrative analysis of single-cell and bulk transcriptomic data has the potential to uncover subtle regulatory shifts in the human lineage underlying novel expression profiles.

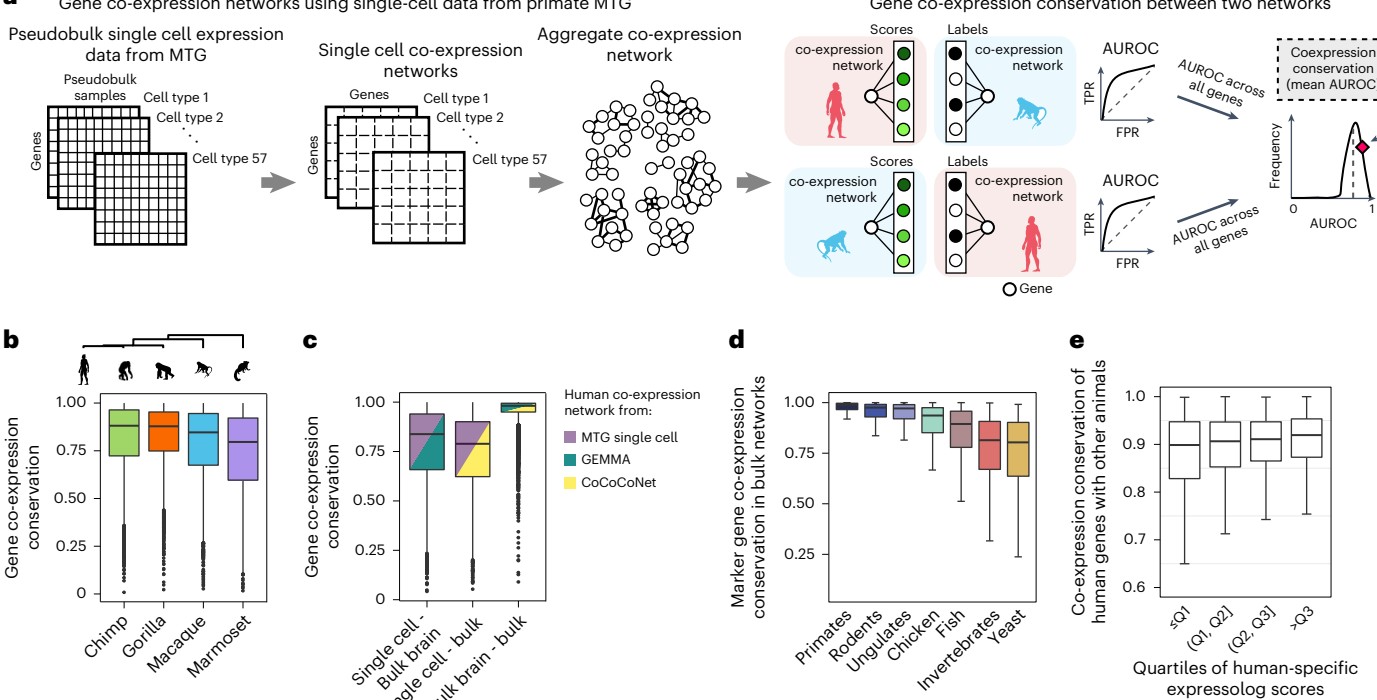

**Fig. 4 | Using co-expression conservation to characterize ancient regulatory landscapes. a**, Aggregate co-expression networks at single-cell resolution built by aggregating 57 cell type-specific co-expression networks in each primate (left). Gene functional conservation is defined as the similarity of co-expression neighbourhoods for 1:1 orthologues across a species pair (right). FPR, false positive rate; TPR, true positive rate. **b**, Boxplots showing the distribution of co-expression conservation for highly variable genes calculated between human and non-human primates ($n = 4,500$ genes). **c**, Boxplots showing the distribution of co-expression conservation for highly variable genes calculated between human-specific networks at different levels of granularity, indicating the replicability of co-expression signatures between 'compositional' (bulk) and 'co-regulatory' (single cell) networks ($n = 4,500$ genes). **d**, Boxplots showing mean co-expression conservation for marker genes between humans and 21 other species, grouped by their divergence time ($n = 1,681$ genes). Co-expression conservation is negatively correlated with phylogenetic distance (Spearman correlation coefficient = $-0.65$, $P < 2.2 \times 10^{-16}$). **e**, Boxplots showing the distribution of co-expression conservation between humans and other species as a function of the expression profile similarity of orthologues shared between human and non-human primates (binned into quartiles (Q) of increasing expressolog scores, $n = 14,131$ genes). For all boxplots, the bounds of the box represent the first and third quartiles, the thick line represents the median and whiskers represent 1.5× the interquartile range.

## Candidate genes for human-specific expression

Genetic variation within species is known to drive regulatory and phenotypic variation across species[12,42]. Under a neutral model of evolution, we expect a similar constraint of evolutionary drift to apply to gene sequences and expression levels within and across species. The evolutionary trajectory of many genes follows this principle, as evidenced by highly conserved expression and co-expression profiles across large evolutionary timescales (Fig. 4e). However, a few outlier genes can have expression changes due to positive selection on specific regulatory variants, lower mutational constraint or environmental differences across species[43]. Here, we propose that an integrative analysis of high-resolution single-nucleus and well-powered bulk transcriptomic data can combine the specificity of expression across cell types with the similarity of co-expression neighbourhood across divergent species to detect human-specific regulatory variation in an evolutionary context.

Our workflow to identify genes with potential human-specific co-expression patterns is illustrated in Fig. 5a. In brief, we filtered the list of 14,131 genes to exclude lowly expressed genes (genes with average expression in the bottom tenth percentile in the primate MTG datasets), yielding a set of 12,742 genes. Next, we selected genes with low expressolog scores (AUROC < 0.55) within one or more classes between human and non-human primates. These 3,383 genes exhibit diverged expression profiles either in humans, non-human primates or across all primates. To shortlist the genes with potential human-specific regulatory divergence, we then examined their co-expression conservation

scores across 19 animals (humans and 18 other animals with >60% human orthologues), and only selected genes showing significantly lower conservation between human and other animals, compared with all other pairs of animals. Given that most genes have highly similar co-expression neighbourhoods even across phylogenetically distant species (average co-expression conservation = 0.89), we recognize that our stringent filter provides a robust but conservative estimate of human-specific divergence. We identified 139 genes with concordant human-specific functional divergence in single-cell and bulk transcriptomic data, a very small fraction of all genes analysed, consistent with an evolutionarily conserved regulatory landscape across species. Genes exhibiting species-specific functional divergence between humans and other primates are the exception in our analysis, not the rule.

Among 3,383 genes with diverged expression profiles in one or more cell classes between human and non-human primates, 98% of genes showed expression divergence in only one cell class, with nearly half of the genes exhibiting differential co-expression across non-neuronal cell types. The 3,383 genes are more likely to be associated with cortex-specific significant expression quantitative trait loci (eQTLs; Wilcoxon $P < 0.003$), which could underlie gene expression changes in humans. Compared with all expressed genes, genes with diverged expression in one or more classes were enriched for intracellular signal transduction, synapse organization and function (Fisher's exact test, adjusted $P < 0.02$), and significantly associated with various

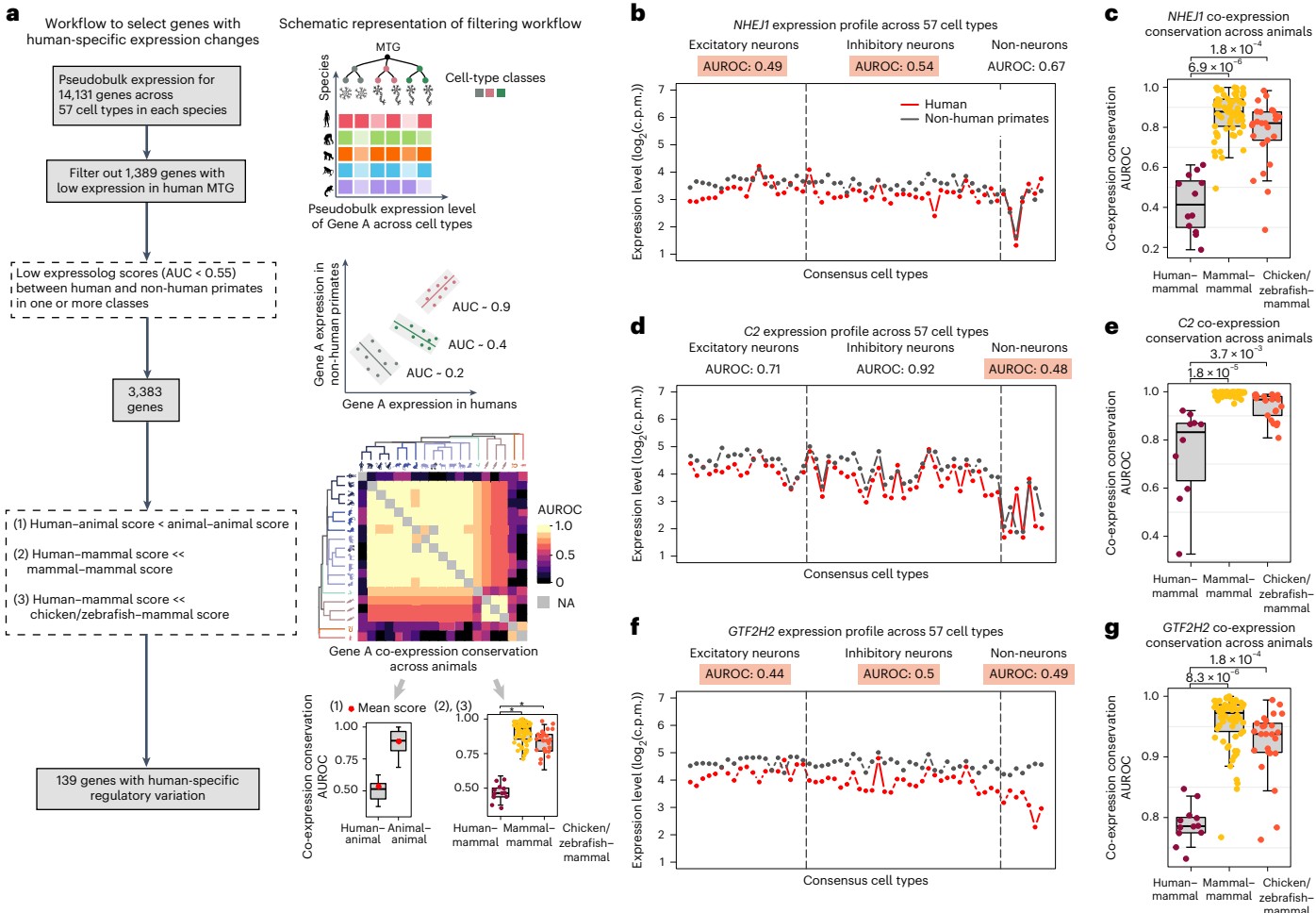

**Fig. 5 | Integrative analysis of single-nucleus and bulk transcriptomic data can detect genes with human-specific regulatory divergence. a**, Schematic representation and workflow of our approach to identify genes with human-specific regulatory changes. AUC, area under the curve. **b**–**g**, Three examples of genes displaying human-specific differential co-expression: *NHEJ1* (**b**,**c**), *C2* (**d**,**e**) and *GTF2H2* (**f**,**g**). **b**,**d**,**f**, Plots comparing the expression profile of each gene of interest in humans with the average expression profile of the orthologue in non-human primates. Expressolog scores within each cell class are listed above the plot, and scores <0.55 are highlighted in orange. **c**,**e**,**g**, Boxplots showing co-expression conservation for orthologues between human and non-human mammals (maroon, *n* = 13 species), between pairs of non-human mammals

(yellow, *n* = 12 species) and between non-human mammals and other vertebrates (chicken and zebrafish; orange, *n* = 14 species). For all boxplots, the bounds of the box represent the first and third quartiles, the thick line represents the median and whiskers represent 1.5× the interquartile range. For each gene, one-sided Wilcoxon test was performed to evaluate whether the co-expression conservation between human and non-human mammals was lower than that between pairs of non-human mammals, and between non-human mammals and other vertebrates. The resulting *P* values were adjusted by applying Benjamini–Hochberg procedure, and the adjusted *P* values are indicated above the boxplots in **c**, **e** and **g**.

brain disorders including intellectual disability, microcephaly, epilepsy and autism spectrum disorders (adjusted *P* < 0.001). We detected 139 genes with putative novel regulatory relationships in humans, and a majority of these genes (68%) were diverged in a single cell class with roughly equal numbers of genes diverged in each of the three classes, GABAergic, glutamatergic and non-neuronal cells. The 139 human genes were also significantly associated with intellectual disability and blindness.

We visualized the expression variation over homologous cell types and cross-species co-expression conservation for three candidate genes showing human-specific deviation in expression profile in neurons (*NHEJ1*), non-neurons (*C2*) and all cell classes (*GTF2H2*). Differences in gene expression profiles between human and non-human primates for these genes are shown in the left-hand panels in Fig. 5b,d,f. The boxplots on the right in Fig. 5c,e,g show the corresponding distributions of co-expression conservation between humans and non-human mammals, within non-human mammals and between non-human mammals

and other model vertebrates (chicken and zebrafish). This broader species analysis confirms differential co-expression in humans, validating the human-specific expression variation observed in single-nucleus transcriptomic data.

*NHEJ1* is a DNA repair gene known to be under positive selection exclusively in the human lineage[44]. An independent study that compiled a comprehensive list of human accelerated regions (HARs) in the genome[45] also identified a HAR overlapping this gene (HARsv2_1598), suggesting accelerated evolution of its coding sequence drives regulatory divergence specific to the human lineage.

Given that *cis*-regulatory variation contributes to interspecific expression divergence[42], association of significant eQTL with *GTF2H2* and *C2* could explain their expression variability across species. *GTF2H2* is a transcription factor gene with high interindividual variability because of several cortex-specific eQTLs, as seen in Genotype-Tissue Expression Project V8 (ref. 46)). *C2* is an immune-related gene involved in interferon signalling and has microglia-specific expression in the

human central nervous system[47]. *C2* is known to mediate interactions between microglia and neurons, and its downregulation in microglia is associated with ageing[48]. Because *C2* has similar expression levels in microglia and neurons, regulatory changes in the human lineage could underlie the divergent pattern of *C2* expression in non-neurons.

Another gene with potential differential expression regulation specific to the human lineage is *BBS5*. We observed human-specific upregulation of *BBS5* specifically in one layer five excitatory neuron cell type and in microglia (Supplementary Fig. 4b), along with differential co-expression conservation between humans and other animals in our bulk networks (Supplementary Fig. 4c). Single-cell epigenomic profiling of broad cortical cell types in the adult and developing human brain[49,50] indicated microglia-specific activity of putative *BBS5* enhancer, strongly suggesting a potential mechanism for expression upregulation specific to human microglia (Supplementary Information). These examples suggest the power of integrative analysis to uncover both patterns and mechanisms of human-specific expression variation, and measure the functional impact in a broad phylogenetic context.

Finally, we sought to assess genic properties of the 139 human genes that could be associated with human-specific functional divergence. Consistent with previous research[51], we found that the 139 genes were younger (Wilcoxon $P < 0.006$), shorter in length (Wilcoxon $P < 3.1 \times 10^{-16}$), had higher GC content (Wilcoxon $P < 2.4 \times 10^{-5}$) and displayed more cell type-specific expression (Wilcoxon $P < 0.0003$) compared with the other 12,603 functionally conserved genes. Divergent genes had marginally lower sequence similarity across primates compared with conserved genes (Wilcoxon $P < 0.01$), coupled with higher sequence evolution rates in the human lineage (Wilcoxon $P < 0.01$). Despite not being more likely to be associated with significant cortical eQTLs, divergent genes showed relatively higher tolerance to inactivation (higher loss-of-function observed/expected upper bound fraction (LOEUF) scores; Wilcoxon $P < 8.4 \times 10^{-8}$). These observations suggest that the divergent genes predominantly evolve under relatively mild evolutionary constraints, with a handful of genes acquiring new regulatory features (like HARs) under positive selection.

## Discussion

Single-nucleus transcriptomic profiling of the MTG in humans and four non-human primates provides an unprecedented opportunity to determine the core transcriptional features underlying conserved cell identity across primates and isolate human-specific transcriptional features related to cellular diversity and trait evolution in the human lineage. In this study, we generated a transcriptomic catalogue of primate MTG cell types, which serves as the basis for comparative analysis of gene expression across primates. Expression profile similarity of 14,131 orthologues between human and non-human primates confirmed the functional conservation of orthologues across primates (mean expressolog AUROC = 0.88; 47% of genes highly conserved with AUROC > 0.95), with the average extent of conservation recapitulating phylogenetic distances.

One of the main goals of comparative analysis using high-resolution, multi-omic profiling of matched brain regions across species is to develop methods for robust inference of genes with human-specific regulatory divergence underlying phenotypic novelty. Given that genes typically have matched expression profiles across primates, differences in co-variation probably reflect functional divergence between species. Therefore, we used cross-species co-expression between human and non-human primates within the three broad classes to identify genes with differential regulation in humans. We found 3,383 genes with divergent transcriptional patterning (expressolog score <0.55) across one or more classes in humans relative to non-human primates. Genes that diverged in one or more classes were significantly associated with multiple neuropsychiatric and neurodegenerative diseases. Most of these genes exhibited

changes in expression profiles only within a single class, and nearly half showed diverged expression limited to non-neuronal cell types. Because gene co-expression reflects shared regulation and function, we verified whether the observed gene expression changes have a functional impact by studying the divergence of gene co-expression relationships across species.

We utilized species-specific gene co-expression networks (generated by large-scale meta-analysis of 49,796 RNA-seq samples spanning 19 animals) to provide a quantitative framework to connect changes in gene expression profile to species-specific differential regulation. We identified 139 genes (<1% of all expressed genes) with human-specific expression and connectivity patterns not replicated in other primates or mammals. Relative to other expressed genes, these 'human-divergent' genes are younger, and display significantly higher rates of sequence evolution and evolve under relaxed mutational constraint. We propose that integrating both types of data can detect both conserved genes that are well-suited for translational research, and genes with differential co-regulation across species that could limit their utility as disease biomarkers in model organisms[52].

Most single-cell comparative studies have focused on differential gene expression across species to isolate species-specific changes in gene activity suggesting functional divergence. However, the functional consequence of differential expression is not always obvious, and the minimal overlap of differentially expressed genes across independent studies further complicates any efforts in meta-analysis to infer robust signatures of species-specific regulatory variability. Although we connect changes in gene expression to changes in co-expression network connectivity to select genes with human-specific regulatory features, we recognize that we underestimate the extent of human-specific functional divergence for several reasons:

(1) Because primates have recently evolved from their last common ancestor, most genes have 1:1 orthologues across species. However, species-specific paralogues are more likely to be divergent across species (Fig. 3F in ref. [28]), but are not in the scope of our current analysis.

(2) Changes in expression profile may be unrelated to differential co-expression, so our strategy of selecting genes with concordant divergence in expression and co-expression patterns misses genes that are functionally diverged due to other sources of variability (like environment or diet).

(3) Our co-expression networks are not MTG- or brain-specific, but built from heterogeneous samples from bulk tissue sequencing. Although we cannot rule out genes with conserved regulation across all tissues except the brain, our current workflow excludes such genes because they have otherwise high co-expression conservation across species.

Although much of our analysis focused on cell types shared by all five primates, we appreciate the importance of species-specific cellular novelty in driving evolutionary change. Although we identified 29 cell types shared across a subset of primates (Supplementary Fig. 1a,c), we are currently underpowered to identify cell types unique to a single species. We hope that our datasets provide a valuable resource to uncover species-specific cell types when aligned against higher resolution spatio-transcriptomic atlases that may become available in the near future.

Overall, we provide a framework to identify genes with marked changes in both cell type-specific expression and gene co-expression neighbourhoods, which could underlie evolutionary innovations exclusive to the human lineage. We make our comprehensive catalogues of single-cell gene expression and cross-metazoa co-expression conservation accessible through a web-based tool (https://gillislab.shinyapps.io/Primate_MTG_coexp/) for users to explore gene functional conservation at single-cell resolution and across large evolutionary distances, and examine the regulatory divergence of genes associated with human-specific traits and diseases. Although our results focused

on genes with human-specific differential regulation, our datasets and framework can be extended to identify genes with differential regulation specific to other species or phylogenetic groups, removing a critical bottleneck in the use of single-cell data and offering exciting opportunities for novel evolutionary analyses of disparate systems.

## Methods

### snRNA-seq processing and clustering

**Cell-type label transfer.** Subclass annotations from the human primary motor cortex (M1) taxonomy[23] were used to annotate cell subclasses in the primate MTG by performing label transfer with Seurat v.3 (ref. [53]). Datasets were preprocessed with the standard LogNormalization method, followed by the selection of 3,000 highly variable genes (or their orthologues for non-human primates) with the 'vst' method, and label transfer based on the first 30 principal components. Each dataset was split into five neighbourhoods (CGE-derived and medial ganglionic eminence (MGE)-derived inhibitory neurons, intratelencephalic (IT) type and deep excitatory neurons, and non-neurons) using the CellSelector (lasso) tool from Seurat to isolate distinct islands of cell populations in the uniform manifold approximation and projection (UMAP) space based on their label transferred identities.

**Filtering low-quality nuclei.** Over 570,000 nuclei were collected from five primates. All nuclei preparations were stained for the pan-neuronal marker NeuN and FACS-purified to enrich for neurons over non-neuronal cells. Samples containing 90% NeuN+ (neurons) and 10% NeuN− (non-neuronal cells) nuclei were used for library preparations and sequencing. SSv4 nuclei were included in downstream analysis if they passed all quality control criteria:

- 30% complementary DNA longer than 400 base pairs;
- 500,000 reads aligned to exonic or intronic sequence;
- 40% of total reads aligned;
- 50% unique reads;
- 0.7 TA nucleotide ratio.

Next, quality control was performed at the neighbourhood level. Neighbourhoods were split into more than 100 metacells using Louvain clustering, and low-quality metacells with relatively low unique molecular identifiers (UMIs) or gene counts (glia and neurons with fewer than 500 and 1,000 genes detected, respectively), predicted doublets (nuclei with doublet scores above 0.3), and/or low subclass label prediction metrics within the neighbourhood (for example, excitatory labelled nuclei that clustered with majority inhibitory or non-neuronal nuclei) were removed from the dataset. Remaining high-quality nuclei were normalized with SCTransform v.1 (ref. [54]) using default parameters.

**snRNA-seq clustering.** Neighbourhoods across individuals and modalities within a species were integrated by identifying mutual nearest-neighbour anchors and applying canonical correlation analysis as implemented in Seurat v.3. The SSv4 dataset (where available) was treated as an individual donor in the integration strategy. For example, deep excitatory neurons from human-Cv3 were split by individuals and integrated with the human-SSv4 deep excitatory neurons. The SelectIntegrationFeatures function was used to identify 3,000 genes for integration, and datasets were integrated across the first 30 principal components. Sex and mitochondrial genes from the gene exclusion list were removed from the list of 3,000 genes used for integration. The gene exclusion list used in this study was derived in Hodge et al.[2], and can be accessed at https://github.com/AllenInstitute/Great_Ape_MTG/blob/master/exclusiongenes_mito_sex_tissue.txt

The integrated space containing the remaining genes was then scaled and projected into 30 principal components, which were used for the clustering of each neighbourhood. Each neighbourhood was clustered using a previously described 'shatter and merge' approach[23]. Louvain clustering was performed using the FindClusters algorithm from Seurat with variable resolution parameters until more than 100 clusters or 'metacells' were identified for each neighbourhood. Metacells were merged with their nearest neighbour until each metacell contained more than 20 nuclei, and had a total of 8 genes (4 for glia) or more differentially expressed with every other metacell. Here, differentially expressed genes are defined as being expressed in more than half of nuclei in both metacells, have a fold-change of two or more across the metacell pair and have a proportion expressed differential of 0.3 or greater. The remaining clusters underwent further quality control to exclude low-quality and outlier populations. These exclusion criteria were based on irregular groupings of metadata features that resided within a cluster. Readers are encouraged to refer to Jorstad et al.[26] for further details on RNA-seq processing, quality control and annotation.

### Replicability of clusters

All analyses were performed in R v.4. MetaNeighbor v.1.12 (refs. [29,30]) was used to provide a measure of neuronal and non-neuronal subclass and cluster replicability within and across species. We used OrthoDB v.10.1 (ref. [32]) to shortlist 14,131 orthologues across five primates, and subset snRNA-seq datasets from each species to this list of common orthologues before further analysis. For each assessment, we identified highly variable genes using the get_variable_genes function from MetaNeighbor. To identify homologous cell types, we used the MetaNeighborUS function with the fast_version and one_vs_best parameters set to TRUE. The one_vs_best parameter identifies highly specific cross-dataset matches by reporting the performance of the closest neighbouring cell type over the second closest as a match for the training cell type, and the results are reported as the relative classification specificity (AUROC). This step identified highly replicable cell types within each species and across each species pair. All 24 subclasses are highly replicable within and across species (one_vs_best AUROC of 0.96 within species and 0.93 across species in Fig. 2b).

Although cell-type clusters are highly replicable within each species (one_vs_best AUROC of 0.93 for neurons and 0.87 for non-neurons), multiple transcriptionally similar clusters mapped to each other across each species pair (average cross-species one_vs_best AUROC of 0.76). To build a consensus cell-type taxonomy across species, we defined a cross-species cluster as a group of clusters that are either reciprocal best hits or clusters with AUROC >0.6 in the one_vs_best mode in at least one pair of species. This lower threshold (AUROC >0.6) reflects the high level of difficulty/specificity of testing only against the best-performing other cell type. To demonstrate the significance of this threshold empirically, we permuted the within-species cluster labels in chimp and mapped them to clusters in the human data by running MetaNeighborUS in the one_vs_best mode. We observed no hits between human and chimp at an AUROC threshold of 0.6, and only one or two hits at a lower threshold of 0.51, highlighting the difficulty in obtaining a uniquely good hit across species.

We identified 86 cross-species clusters, each containing clusters from at least two primates. Individual clusters that could not be uniquely mapped to a single cross-species cluster were assigned to one of the 86 cross-species clusters based on their transcriptional similarity. For each such cluster, top ten of their closest neighbours were identified using MetaNeighborUS one_vs_all cluster replicability scores, and the cluster was assigned to the cross-species cluster in which a strict majority of its nearest neighbours belong. For clusters with multiple best hits, this was repeated using top 20 closest neighbours, still requiring a strict majority to assign a cross-species type. A total of 594 clusters present in all five primates mapped to 86 cross-species clusters, with 492 clusters present in 57 consensus cross-species clusters shared by all five primates. Five of the 57 consensus cell types are visualized in the third panel in Fig. 2c. Gene expression across single nuclei present

in 57 consensus clusters in the human MTG was visualized using UMAP plots coloured by log-transformed expression levels (Fig. 3c).

## Calculating the expressolog score

We generated a pseudobulk dataset for each species that recorded the normalized average counts per cell type of 14,131 genes across the consensus cell types. For each pair of species, we calculated the expression profile similarity for all pairs of genes by computing the Pearson correlation of normalized expression levels across 57 homologous cell types. For each gene in one species, we calculated the rank-standardized expression profile similarity of its 1:1 orthologue (relative to 14,130 genes) in the other species, repeated this calculation in the opposite direction, and report the average of the bidirectional scores as the 'expressolog score' (Fig. 3b). The expressolog score is equivalent to the average AUROC, with a score of 1 indicating that orthologues can be identified by matching expression profiles across species, and a score of 0 suggesting that orthologues have diverged in expression across species. The AUROC for expression profile similarity of gene $i$ in one species with gene $j$ in another species was calculated as:

$$\text{AUROC} = (r_{ij} - 1)/(N - 1),$$

where $N$ is the total number of genes, and $r_{ij}$ is the rank of the Pearson correlation of gene $i$ with gene $j$ relative to other $(N - 1)$ genes in the second species. In our expressolog analysis, $N = 14,131$.

Expressolog scores were also calculated across cell types within each class, subtype and meta-cluster (as defined in Fig. 3a) using the same formula, and capture the extent to which gene expression variation within progressively homogeneous cell types is shared across species. For each cell-type group (excitatory neurons, medial ganglionic eminence (MGE)-derived inhibitory neurons or *Pvalb* meta-cluster), we obtained a 14,131 × 14,131 matrix of AUROCs corresponding to the expression profile similarity of all gene pairs across a pair of species, and report the AUROCs corresponding to 1:1 orthologues as expressolog scores. Average expressolog scores between human and non-human primates calculated within each cell-type group are reported in Supplementary Table 3.

## Isoform data generation

We used SSv4 snRNA-seq data from human, chimp and gorilla to assess the expression profile similarity of individual isoforms across great apes. Reads from cells belonging to the consensus clusters were mapped to the species' genomes using the default parameters in STAR v.2.7.7a (ref. 55). Isoform and gene expression were quantified using RSEM v.1.3.3. For the analysis related to Fig. 3, we retained consensus clusters with reads mapped from ten or more cells, and further removed isoforms with total expression <5 transcripts per million. To assess whether an isoform could predict itself among other isoforms of a gene, we considered genes with at least two isoforms shared by all species. We computed the expressolog scores of all pairs of isoforms of a gene across a pair of species, and ranked the expressolog score of an isoform with itself relative to other isoforms (reported as an AUROC).

## Building aggregate co-expression networks

All co-expression networks used in this study were generated by aggregating networks built from individual cell types or datasets. For each consensus cell type, expression data were filtered to the set of 4,500 highly variable genes between human and non-human primates, and randomly split into samples of 20 nuclei each. The read counts were aggregated across 20 nuclei in each sample. The corresponding co-expression network was built by calculating the Spearman correlation between all pairs of highly variable genes across these pseudobulk samples, and then ranking the correlation coefficients for all gene–gene pairs, with NAs (not available) assigned the median rank. An aggregate

single-cell co-expression network for each primate was generated by averaging the rank-standardized networks from individual cell types. For example, the human single-cell co-expression network was built by averaging the 57 cell type-specific networks (human|Astro_1, human|Oligo_9, human|Lamp5_2, human|Sst_20, human|L4 IT_10, human|L5 ET_1, and so on).

Meta-analytic bulk co-expression networks for 21 metazoan species and yeast derived by aggregating 54,668 individual RNA-seq datasets in a similar manner were downloaded from CoCoCoNet[27]. Four RNA-seq datasets each were used to build aggregate co-expression networks for gorilla and marmoset. Human bulk brain co-expression network was generated by aggregating 20 individual datasets curated by Gemma[39].

## Curated gene sets and orthology

To investigate the conservation and divergence of the co-expression of gene families between human and non-human primates, we carried out MetaNeighbor analysis using gene groups curated by HGNC at the European Bioinformatics Institute (https://www.genenames.org; downloaded October 2021) and by SynGO[56] (downloaded October 2021). HGNC annotations were propagated via the provided group hierarchy to ensure the comprehensiveness of parent annotations. Only groups containing five or more genes were included in the analysis.

The MetaMarkers package[57] was used to find marker genes for cell types defined at different levels of organization in each species, with the search at each level stratified by the broader cell type to generate marker sets that can discriminate even relatively homogeneous cell clusters. Marker genes defining cell class, subclass and consensus clusters are listed in Supplementary Table 4. A list of transcription factors used in Fig. 3 was obtained from Ziffra et al.[50].

To assess genic features associated with human-divergent genes, we downloaded the sequence similarity, gene length and GC content for all human genes from Ensembl v.107, gene ages from GenTree (http://gentree.ioz.ac.cn/), list of significant eQTLs and associated genes from the Genotype-Tissue Expression Project V8 portal[46], and gene constraint scores (loss-of-function observed/expected upper bound fraction) from The Genome Aggregation Database (gnomAD v.2.1.1). Average sequence evolution rates between human and other primates, and gene lists associated with various brain disorders were downloaded from the GenEvo website (https://genevo.pasteur.fr/). Gene set enrichment analysis was performed using Fisher's exact test and the resulting $P$ values were adjusted by applying Benjamini–Hochberg correction. Cell type-specificity scores were calculated using pseudobulk human MTG data as published[58].

OrthoDB v.10.1 (ref. 32) was used for orthology mapping. For each pair of species, we used the set of orthology groups of their last common ancestor to obtain a comprehensive list of many-to-many orthologues. We filtered this list to include only 1:1 orthologues, which yielded ~4,500 orthologues for phylogenetically distant species (like human and yeast) and ~13,500 orthologues for recently diverged species. All single-nucleus expression profile similarity analyses used a set of 14,131 orthologues across five primates, with aggregate co-expression networks built using a subset of the top 4,500 highly variable genes. Species divergence times were sourced from TimeTree[59].

## Calculating cross-species co-expression conservation

For each pair of species to be compared, we filtered aggregate co-expression networks to include known 1:1 orthologous genes, then compare each gene's top ten co-expression partners across species to quantify gene functional similarity[28]. The scheme in Fig. 4a illustrates the calculation of co-expression conservation of an orthologous gene between human and rhesus macaque. Given a gene of interest in human, its top ten co-expression partners are identified and co-expression conservation is calculated by ranking the

co-expression of their macaque orthologues with the macaque orthologue of the target gene. Calculation is repeated in the other direction (macaque to human), and the average of the bidirectional AUROC is taken as a measure of co-expression neighbourhood similarity of the target gene. We calculate the co-expression conservation not just for orthologues, but for all gene pairs, and rank the co-expression conservation of each orthologue relative to all genes to determine the specificity of co-expression neighbourhood conservation for each gene. We term this specificity score 'co-expression conservation' and note that it provides a standardized measure to compare the extent of functional conservation of orthologues over large evolutionary timescales, and infer examples of human-specific regulatory divergence.

Co-expression conservation for a set of 4,500 highly variable genes shared across human single-cell, bulk brain and bulk co-expression networks was used to assess the topological similarity across the different networks (Fig. 4c). We use a set of 1,681 human markers (comprising 582 class, 929 subclass and 170 consensus cell-type markers listed in Supplementary Table 4) to assess the extent of functional conservation across metazoa. We compute the co-expression conservation of 1,681 marker genes between human and 21 other metazoan species using aggregate co-expression networks derived from bulk transcriptomic data, and observe that marker genes have conserved co-expression neighbourhoods even across phylogenetically distant species (Fig. 4d). Classification of 14,131 genes based on their co-expression divergence in single-cell, bulk or both transcriptomic datasets is provided in Supplementary Table 6.

### Protein sequence similarity for candidate genes showing regulatory divergence in humans

We obtained data for the protein sequence similarity of 1:1 orthologues between humans and non-human primates from Ensembl v.107. *NHEJ1* showed 96% and *GTF2H2* showed 91% average similarity between human and non-human primates (great apes: chimp, gorilla; monkeys: crab-eating macaque, rhesus macaque). *C2* had 97.87% similarity between humans and crab-eating macaque. *BBS5* showed 99.7% similarity between human and two great apes.

### Impact of different subclass annotation protocols on gene expression profile similarity

To test the robustness of our results to different cell subclass annotation strategies, we re-annotated the primate MTG datasets using the human MTG taxonomy described in Hodge et al.[2]. Briefly, we used an automated method (MetaMarkers[57]) to generate marker gene sets for the 22 cell subclasses observed in Hodge et al.[2], which were then used to annotate cell types at the subclass level for all primate MTG datasets. Cell subclass annotations for cell-type clusters reported in Hodge et al.[2], and their corresponding marker gene sets can be accessed at https://labshare.cshl.edu/shares/gillislab/resource/Primate_MTG_coexp/Hodge_MTG_subclass_anno_marker_list.xlsx.

We found that the subclass annotations generated using the MTG[2] and M1 (ref. 26) taxonomies were mostly concordant (overall classification accuracy = 0.9, and adjusted Rand index = 0.79). Next, we recalculated the expressolog scores for 14,131 genes at the subclass level (expression profile similarity of 1:1 orthologues across 22 cell types for each pair of primates), and compared them with the subclass-level expressolog scores calculated using subclass annotations provided by Jorstad et al.[26]. We observed that the average expressolog scores mirrored species divergence times, and were largely independent of subclass annotation strategy (Supplementary Fig. 7a,b). Comparison of the expressolog scores between human and non-human primates for the 139 human-specific 'diverged' genes (Fig. 5a) and the remaining 'conserved' genes indicated that the genes diverged in humans had significantly lower expression profile similarity with their non-human primate orthologues irrespective of the subclass labelling method (Supplementary Fig. 7c).

### Reporting summary

Further information on research design is available in the Nature Portfolio Reporting Summary linked to this article.

### Data availability

Raw sequence data were produced as part of the BRAIN Initiative Cell Census Network (BICCN) and are available for download from the Neuroscience Multi-omics Archive (https://assets.nemoarchive.org/dat-net1412), and the relevant file locations are listed in https://github.com/hamsinisuresh/Primate-MTG-coexpression. Integrated snRNA-seq gene expression dataset and associated metadata for each primate species are available on the BICCN Human/NHP website. Data used in the analyses reported in this study can be accessed from https://labshare.cshl.edu/shares/gillislab/resource/Primate_MTG_coexp/ and from the figshare repository[60]: https://doi.org/10.6084/m9.figshare.22032104.

### Code availability

Details on the processing and clustering of primate MTG snRNA-seq datasets are available at: https://github.com/AllenInstitute/Great_Ape_MTG. Code to reproduce the cross-species expression profile similarity and co-expression conservation analysis can be accessed from https://github.com/hamsinisuresh/Primate-MTG-coexpression.

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

## Acknowledgements

This publication was supported by and coordinated through the BRAIN Initiative Cell Census Network. This publication is part of the Human Cell Atlas- www.humancellatlas.org/publications/. We thank L. French, S. Fischer, R. Kawaguchi, M. Passalacqua and J. Hover for thoughtful feedback on the manuscript. This work was supported by National Institutes of Health (NIH) grant nos R01LM012736, R01MH113005, U19MH114821 (H.S. and J.G.), NIH grant no. F32MH114501 and National Association for Research on Schizophrenia and Depression Young Investigator Award (M.C.), NIH grant no. R01HG009318 (A.D.) and NIH grant no. U01MH114812 (N.J., R.H., E.L. and T.B.).

## Author contributions

N.J., R.H., E.L. and T.B. prepared the samples and generated RNA data. A.D. generated the isoform data. J.G. conceptualized the study. H.S. analysed the data. H.S., M.C., A.D. and J.G. interpreted the data. H.S., M.C. and J.G. wrote the manuscript.

## Competing interests

From 11 April 2022, N.J. is an employee of Genentech. The remaining authors declare no competing interests.

## Additional information

**Correspondence and requests for materials** should be addressed to Jesse Gillis.

# Reporting Summary

## Statistics

For all statistical analyses, confirm that the following items are present in the figure legend, table legend, main text, or Methods section.

| n/a | Confirmed | |
|---|---|---|
| ☐ | ☒ | The exact sample size (*n*) for each experimental group/condition, given as a discrete number and unit of measurement |
| ☐ | ☒ | A statement on whether measurements were taken from distinct samples or whether the same sample was measured repeatedly |
| ☐ | ☒ | The statistical test(s) used AND whether they are one- or two-sided<br>*Only common tests should be described solely by name; describe more complex techniques in the Methods section.* |
| ☒ | ☐ | A description of all covariates tested |
| ☐ | ☒ | A description of any assumptions or corrections, such as tests of normality and adjustment for multiple comparisons |
| ☐ | ☒ | A full description of the statistical parameters including central tendency (e.g. means) or other basic estimates (e.g. regression coefficient) AND variation (e.g. standard deviation) or associated estimates of uncertainty (e.g. confidence intervals) |
| ☐ | ☒ | For null hypothesis testing, the test statistic (e.g. *F*, *t*, *r*) with confidence intervals, effect sizes, degrees of freedom and *P* value noted<br>*Give P values as exact values whenever suitable.* |
| ☒ | ☐ | For Bayesian analysis, information on the choice of priors and Markov chain Monte Carlo settings |
| ☒ | ☐ | For hierarchical and complex designs, identification of the appropriate level for tests and full reporting of outcomes |
| ☐ | ☒ | Estimates of effect sizes (e.g. Cohen's *d*, Pearson's *r*), indicating how they were calculated |

*Our web collection on statistics for biologists contains articles on many of the points above.*

## Software and code

Policy information about availability of computer code

| | |
|---|---|
| Data collection | No software was used to collect the data in this study |
| Data analysis | R version 4, MetaNeighbor version 1.12, MetaMarkers version 0.0.1, STAR version 2.7.7a, RSEM version 1.3.3 |

For manuscripts utilizing custom algorithms or software that are central to the research but not yet described in published literature, software must be made available to editors and reviewers. We strongly encourage code deposition in a community repository (e.g. GitHub). See the Nature Portfolio guidelines for submitting code & software for further information.

## Data

Policy information about availability of data

All manuscripts must include a data availability statement. This statement should provide the following information, where applicable:
- Accession codes, unique identifiers, or web links for publicly available datasets
- A description of any restrictions on data availability
- For clinical datasets or third party data, please ensure that the statement adheres to our policy

Raw sequence data were produced as part of the BRAIN Initiative Cell Census Network and are available for download from the Neuroscience Multi-omics Archive (https://assets.nemoarchive.org/dat-net1412), and the relevant file locations are listed in https://github.com/hamsinisuresh/Primate-MTG-coexpression. Integrated snRNA-seq gene expression dataset and associated metadata for each primate species are available on the BICCN Human/NHP website. Data used in the analyses reported in this study can be accessed from https://labshare.cshl.edu/shares/gillislab/resource/Primate_MTG_coexp/, and from Figshare (DOI: https://

## Human research participants

Policy information about studies involving human research participants and Sex and Gender in Research.

| | |
|---|---|
| Reporting on sex and gender | N/A |
| Population characteristics | N/A |
| Recruitment | N/A |
| Ethics oversight | N/A |

Note that full information on the approval of the study protocol must also be provided in the manuscript.

# Field-specific reporting

Please select the one below that is the best fit for your research. If you are not sure, read the appropriate sections before making your selection.

☒ Life sciences  ☐ Behavioural & social sciences  ☐ Ecological, evolutionary & environmental sciences

For a reference copy of the document with all sections, see nature.com/documents/nr-reporting-summary-flat.pdf

# Life sciences study design

All studies must disclose on these points even when the disclosure is negative.

| | |
|---|---|
| Sample size | Sample size was not predetermined. To generate the primate MTG atlas, single nuclei were isolated from postmortem brains of human (n = 7), chimpanzee (n = 7), gorilla (n = 4), macaque (n = 3), and marmoset (n = 3). This allowed us to collect over 570,000 nuclei from high quality specimens across five primates that passed stringent quality control, and generated consistent transcriptomic clusters across donors. |
| Data exclusions | No data were excluded from the analysis |
| Replication | Cluster replicability within and across species was assessed using MetaNeighbor. Cell type clusters within each species were highly reproducible across donors and sequencing technologies. Cell types defined at the class, subclass, and consensus cluster levels were highly replicable across primates. |
| Randomization | Randomization is not possible since the categorical variable of interest is "species" |
| Blinding | Blinding of samples is not possible since species-specific properties need to be accounted for in alignment and comparative transcriptomic analysis |

# Reporting for specific materials, systems and methods

We require information from authors about some types of materials, experimental systems and methods used in many studies. Here, indicate whether each material, system or method listed is relevant to your study. If you are not sure if a list item applies to your research, read the appropriate section before selecting a response.

### Materials & experimental systems

| n/a | Involved in the study |
|---|---|
| ☒ | ☐ Antibodies |
| ☒ | ☐ Eukaryotic cell lines |
| ☒ | ☐ Palaeontology and archaeology |
| ☒ | ☐ Animals and other organisms |
| ☒ | ☐ Clinical data |
| ☒ | ☐ Dual use research of concern |

### Methods

| n/a | Involved in the study |
|---|---|
| ☒ | ☐ ChIP-seq |
| ☒ | ☐ Flow cytometry |
| ☒ | ☐ MRI-based neuroimaging |

