## [Peer Review File · Nature Ecology & Evolution]

Peer Review Information

Journal: Nature Ecology & Evolution

Manuscript Title: Comparative single cell transcriptomic analysis of primate brains highlights human-specific regulatory evolution

Corresponding author name(s): Jesse Gillis

Editorial Notes:

Reviewer Comments & Decisions:

Decision Letter, initial version:

23rd March 2023

Dear Jesse,

Your manuscript entitled "Comparative single cell transcriptomic analysis of primate brains highlights human-specific regulatory evolution" has now been seen by three reviewers, whose comments are attached. The reviewers have raised a number of concerns which will need to be addressed before we can offer publication in Nature Ecology & Evolution. We will therefore need to see your responses to the criticisms raised and to some editorial concerns, along with a revised manuscript, before we can reach a final decision regarding publication.

We therefore invite you to revise your manuscript taking into account all reviewer and editor comments. Please highlight all changes in the manuscript text file in Microsoft Word format.

* If you have not done so already please begin to revise your manuscript so that it conforms to our Article format instructions at <http://www.nature.com/natecolevol/info/final-submission>. Refer also to any guidelines provided in this letter.

2[REDACTED]

Nature Ecology & Evolution is committed to improving transparency in authorship. As part of our efforts in this direction, we are now requesting that all authors identified as 'corresponding author' on published papers create and link their Open Researcher and Contributor Identifier (ORCID) with their account on the Manuscript Tracking System (MTS), prior to acceptance. ORCID helps the scientific community achieve unambiguous attribution of all scholarly contributions. You can create and link your ORCID from the home page of the MTS by clicking on 'Modify my Springer Nature account'. For more information please visit www.springernature.com/orcid.

[REDACTED]

Reviewer expertise:

Reviewer #1: computational analysis including coexpression and single cell data

Reviewer #2: human brain, computational analysis including coexpression and single cell data

Reviewer #3: evolution vertebrate brain, single cell genomics

Reviewers' comments:

Reviewer #1 (Remarks to the Author):

Authors use expression patterns within (and between) cell types in MTG among humans and four other primates to identify human specific gene regulation evolution. Authors analyse a single cell MTG dataset and identify common cell types between the five primates. Between these cell types, they identify genes with variable expression patterns among the primates. They use metrics expressolog

2and coexpression conservation (the former previously published and the latter defined in this work) to identify ortholog genes with the most similar expression patterns and measure the variation of their expression patterns among the cell types for each gene among species. Based on these analyses, authors identify genes with human-specific regulatory divergence. This is a great and substantial body of work and the authors seem to be well experienced in working with this data and the various analyses used in this study. However, the text lacks cohesion and details are lacking in some parts, clarification is required in some parts of results and methods sections. Figure captions are mostly minimal as well, making it challenging to understand the analyses and outcomes, or how they fit into the results.

In general, please try to refer to the relevant method section for various measurements and analyses. Please consider describing all the analyses clearly. Please consider explaining what various elements in the figures refer to. In case of the plots, please explain what is being demonstrated and what is inferred. Please refer to the following comments as examples, and modify the text accordingly in other parts as well.

Figure 1A: Please consider explaining the elements in the figure. For example, what do the colour bars refer to?

Figure 1B: Please consider explaining what different elements in the figure represent. For example, what are the lines in the left figure?

Figure 2C: Please mention what are the dimensions in each of the plots.

Figure 2E, B: In the text, when referring to these Figures, please refer to the methods section or citation used for measuring the reproducibility. I think the methods section is "Replicability of clusters".

Figure 5A: inconsistency for count of genes for pseudo-bulk expression in the diagram and the methods section "Calculating the expressolog score".

Major comment on result section "Assessing evolutionary conservation of gene activity using coexpression networks":

Based on the description in "Calculating expression score", the pseudo-bulk samples mentioned in the first paragraph seem to refer to a set of 57 samples (count of consensus cell types) where each gene has the normalised average CPM for that cell type. However, it remains unclear how this dataset can be used to build the aggregate cell type specific networks mentioned in the first paragraph. It also doesn't seem to fit anywhere in the description in the methods section "building aggregate coexpression networks". In Figure 4A, "Pseudobulk single cell" expression data" is mentioned, where samples for each cell type are called "pseudocells". This term is not defined in the text, it is unclear what these samples refer to. The methods section "building aggregate networks" seems to lack many details. For example, it is mentioned that the data for each consensus cell type is randomly split into samples of 20 cells each, and that the corresponding coexpression network is built based on "aggregate read counts". Does this mean that the 20 cell splits were added up and one coexpression network was built from the added samples? If so, please clarify this in the text.

Based on the above notes, the exact process of building these networks is unclear to me, however it seems that these networks are built to reflect the similarity of gene-gene expression patterns among cell types for each of the species.

It is not clear how paragraph 3 ("Are gene coexpression relationships...") relates to other parts of this section or to what the title implies. Paragraph 4 ("While the neocortex...") seems to have started abruptly.

Regarding the coexpression conservation between different networks (Figure 4C), please elaborate on how these measurements can be interpreted.

Regarding the highly variable genes, how was the count 4500 selected for the coexpression analysis, and why is the count 3000 selected for the subclass label transfer?

Minor comment: Inconsistency in compound words - Figure 4A, 5A, "pseudobulk" and "pseudocell" vs. "pseudo-bulk" in the text.

Reviewer #2 (Remarks to the Author):

In this study, Suresh et al. analyze single nuclei from the middle temporal gyrus (MTG) of humans, chimps, gorillas, macaques, and marmosets to investigate evolutionarily conserved and divergent patterns of gene expression. The authors report 57 homologous cell types across the species, in which gene expression patterns are overwhelmingly conserved. Notably, a subset of genes exhibit human-specific patterns, evidenced by divergent signatures across one or more cell classes in single-nucleus data, and divergent regulatory relationships in cross-species bulk gene expression coexpression networks. Divergent genes were characterized as having higher loss-of-function tolerance compared to conserved genes, leading the authors to conclude that human-specific features in the MTG evolved under comparatively mild evolutionary constraints. Finally, the authors point to human-specific gene expression patterns in a disease-implicated gene (BBS5) as a potential source of phenotypic disparity between humans and animal models, thereby demonstrating the broader utility of their approach to predict translational success (or lack thereof) in animal models of disease.

The authors are studying an interesting topic using relatively new data modalities and have developed an impressive track record in this field. Figures are nicely prepared, but analytical threads are hard to follow.

Major comments:

1) This study includes a large amount of data and many complex analyses. The authors need to provide more details in the methods and / or elsewhere to ensure that these analyses are clear and reproducible. In its current form, it is difficult to precisely understand all of the steps that were taken between raw data and the identification of 57 homologous cell types. Breaking down one paragraph (p. 22) as an example:

4"Datasets from each species and modality (SSv4, Cv3) were analyzed independently. Briefly, each dataset was subset into 5 neighborhoods (CGE-derived and MGE-derived inhibitory neurons, IT type and deep excitatory neurons, and non-neurons) based on prior knowledge from human and mouse studies of cortical cell types (2, 23)."

What prior knowledge and how was this done?

"Each neighborhood was annotated using the label transfer function from Seurat (59) with cell subclass labels from the recently published human primary motor cortex (M1) taxonomy (23)."

Does the label transfer function have any parameters? What is the justification for using cell subclass labels from an entirely different brain region (M1) to annotate MTG and why is this not mentioned in the Results?

"Subclass label transfer was performed using 3000 highly variable genes or their orthologs for human and non-human primate datasets, respectively."

How exactly were these 3000 highly variable genes defined? Are they listed anywhere?

"Datasets underwent additional QC and passing nuclei from each dataset were normalized using SCTransform (60)."

What kind of QC and how was this done? Any normalization parameters?

"An integrated space was generated for each species by performing a canonical correlation analysis (CCA) across individuals and modalities. Each integrated space was clustered into hundreds of 'metacells', and metacells which passed quality control were merged with their nearest neighbors until merging criteria were met, resulting in the final clusters for each species (refer to Jorstad et al (26) for further details on RNA-seq processing, QC and annotation)."

How was CCA performed? How was clustering performed? What type of cluster QC was performed? What were the merging criteria? Referring the reader to a dense bioRxiv manuscript for further details is not very helpful. It does not seem that the code underlying the steps described above has been made available in the GitHub repo or elsewhere, but please correct me if I am wrong.

2) What is the relationship between AUROC scores, statistical significance, and multiple testing? If AUROC = 0.5 corresponds to chance, can one establish the 'significance' of arbitrary AUROC thresholds (e.g., 0.6 used for cross-species clustering) based on empirical null distributions of AUROC using permuted data, or otherwise? How is this impacted by multiple testing?

3) All conclusions from the study are derived from the initial identification of homologous cell types, and therefore showing the existence of cell type replicability is paramount. While the authors take steps to validate cell types (leave-one-out cross-validation, MetaNeighbor), the study could benefit from either: i) further demonstrating the presence of homologous cell types in independent datasets

5from the same brain region that were not used in the study, or ii) histological validation of species-specific gene coexpression relationships.

4) Although the motivation for their analysis is clear, the most obvious evidence for evolutionary change may come from the things that are hardest to compare, i.e., putative cell types that are present in one species but not others. Is there anything to be learned by studying the transcriptional phenotypes of non-homologous cell types?

Reviewer #3 (Remarks to the Author):

In the manuscript "Comparative single cell transcriptomic analysis of primate brains highlights human-specific regulatory evolution", Suresh and colleagues present an analysis of single-cell RNA sequencing (scRNAseq) data from the cerebral cortex of five primate species. The aim of the analysis is to identify genes with human-specific expression or co-expression profiles at the cell type level.

This manuscript reads as an extension of Jorstad et al (reference 26), a bioRxiv preprint that describes in detail the five new primate scRNAseq datasets including an integrated cell type taxonomy (the authors of this paper contributed to Jorstad et al).

Complementing Jorstad et al, this manuscript focuses on the evolution of gene coexpression. The authors identify 57 homologous cell types in primates. They then build coexpression networks from bulk RNA sequencing data and analyze network connectivity of the human-specific genes in the 57 homologous cell types. They find evidence for relaxed selective constraints for a small set of genes with human-specific cell type expression.

This manuscript will be relevant for readers interested in the evolution of gene expression patterns, cell types, and human-specific traits.

The manuscript is well written. I appreciated the figure schematics illustrating the logic of every data analysis step; this really helps navigate the paper.

From a bioinformatic and statistical perspective, the analysis seems sound and is well documented in the methods sections (although parts of the analysis are outside my immediate field of expertise). Integrating single-cell and bulk RNAseq data is an interesting approach, and applying this approach to other contexts may lead to biological insights. The paragraph in the Discussion outlining the limitations of the study is extremely valuable.

My major point of concern is the interpretability and significance of the gene coexpression network analysis (figures 4 and 5).

Figure 4

Figure 4D reports that coexpression scores of 1:1 orthologs are conserved in metazoa. This is surprising and puzzling, because it implies that animals with a wide variety and divergence of cell

6types still share a large number of gene expression modules. I wonder whether the extent of conservation is overestimated because of the way the analysis is set up:

1. Selection of marker genes

What marker genes were included in the 1:1 orthologs set used for figure 4? How are they expressed across the primate cell type dataset? If the majority of the markers distinguish neurons vs non-neuronal cells, for example, then the results in figure 4 would be less surprising (enrichment of pan-neuronal specific coexpression sets).

How sensitive are these results to the set of marker genes - in other words, do these results change significantly if the criteria for marker gene selection change? Does this set show significant enrichment for genes in specific functional categories? Anything specific to point out about the outliers in fig 4B? (For example: In what cell types are these expressed? Do they belong to specific functional categories?)

2. Tissue heterogeneity in co-expression networks

A critical issue may lie in the fact that the coexpression networks are built from bulk RNAseq data from a variety of tissues, not just brain. There are many examples of genes defining neuronal identity (example: transcription factors, extracellular matrix) that are co-expressed with completely different genes in different tissues, for example brain and muscle. To my understanding (but I might be wrong), those genes would get a low coexpression conservation score in this analysis, because the coexpression networks are built from a heterogeneous collection of tissues. Would it be possible to repeat the analysis using networks built on bulk brain data only?

Figure 5:

As the authors point out, the analysis underlying identification of the 139 genes with human-specific regulatory divergence is extremely conservative, and may underestimate the number of genes with human-specific divergence. This analysis was obviously human-centric. What would we see with, say, a gorilla-centric, or a marmoset-centric analysis? If analyses centered on other primate species were to yield a similar number of diverging genes, that would suggest that humans are not that special after all.

*****END*****

Author Rebuttal to Initial comments

Reviewer #1 (Remarks to the Author):

Authors use expression patterns within (and between) cell types in MTG among humans and four other primates to identify human specific gene regulation evolution. Authors analyse a single cell MTG dataset and identify common cell types between the five primates. Between these cell types, they identify genes with variable expression patterns among the primates. They use metrics expressolog and coexpression conservation (the former previously published and the latter defined in this work) to identify ortholog genes with the most similar expression patterns and measure the variation of their expression patterns among the cell types for each gene among species. Based on these analyses, authors identify genes with human-specific regulatory divergence. This is a great and substantial body of work and the authors seem to be well experienced in working with this data and the various analyses used in this study. However, the text lacks cohesion and details are lacking in some parts, clarification is required in some parts of results and methods sections. Figure captions are mostly minimal as well, making it challenging to understand the analyses and outcomes, or how they fit into the results.

In general, please try to refer to the relevant method section for various measurements and analyses. Please consider describing all the analyses clearly. Please consider explaining what various elements in the figures refer to. In case of the plots, please explain what is being demonstrated and what is inferred. Please refer to the following comments as examples, and modify the text accordingly in other parts as well.

We thank the reviewer for their detailed feedback on the manuscript, and we have now edited the text for enhanced clarity.

Figure 1A: Please consider explaining the elements in the figure. For example, what do the colour bars refer to?

Thank you for pointing this out. We have edited the figure and caption to explain the various elements in this figure panel.

"Bar plot shows the percentage of within-species cell type clusters associated with each consensus cell type, colored by species."

Figure 1B: Please consider explaining what different elements in the figure represent. For example, what are the lines in the left figure?

We have edited the figure caption to explain the various elements in Fig 1B.

"(left) We quantify the similarity of gene expression profiles across primates to select genes with conserved expression signatures across non-human primates but diverged in humans."

“(right) Schematic indicates that the gene retains its top 10 coexpression partners in all animals except humans, suggesting differential coexpression connectivity could underlie human-specific expression divergence.”

Figure 2C: Please mention what are the dimensions in each of the plots.

We have edited the figure to include dimensions in each of the plots (where applicable). We assess the replicability of cell types across each species pair using MetaNeighbor. For each pair of primates, the SCTransform-normalized counts matrices are given as inputs, and MetaNeighbor provides two cluster replicability score matrices, each having dimensions of (# within-species clusters in primate 1) x (# within-species clusters in primate 2). The cluster replicability scores in the top panel are obtained from running MetaNeighborUS function with one_vs_best parameter set to TRUE. In this mode, cluster replicability scores (AUROCs) are computed between the two closest neighbors in the test dataset, where the closer neighbor will have the higher score, and all others are shown in gray (NA). The cluster replicability scores in the bottom panel are obtained by running MetaNeighborUS function with one_vs_all parameter set to TRUE, where AUROCs are computed between all pairs of clusters.

Figure 2E, B: In the text, when referring to these Figures, please refer to the methods section or citation used for measuring the reproducibility. I think the methods section is “Replicability of clusters”.

Thank you for this suggestion, we have made this change.

Figure 5A: inconsistency for count of genes for pseudo-bulk expression in the diagram and the methods section “Calculating the expressolog score”.

We apologize for not explaining the discrepancy in gene counts used to calculate expressologs in Fig 3 and those used to select human-divergent genes in Fig 5. Although we calculate the expressolog score for all 14,131 genes, we filtered this list further so that the inferred patterns of human-specific expression divergence are not influenced by low expression of genes in one or more primates. Specifically, we removed 1,389 genes with average expression in the bottom 10 percentile in human and non-human primate datasets. We have edited the workflow in Fig 5A and text in the Results section “Genes with human-specific regulatory divergence are associated with relaxed evolutionary constraint” to reflect this additional filtering step.

Major comment on result section “Assessing evolutionary conservation of gene activity using coexpression networks”:

Based on the description in “Calculating expression score”, the pseudo-bulk samples mentioned in the first paragraph seem to refer to a set of 57 samples (count of consensus cell types) where each gene has the normalised average CPM for that cell type. However, it remains unclear how this dataset can be used to build the aggregate cell type specific networks mentioned in the first paragraph. It also doesn't seem to fit anywhere in the description in the methods section

"building aggregate coexpression networks". In Figure 4A, "Pseudobulk single cell" expression data" is mentioned, where samples for each cell type are called "pseudocells". This term is not defined in the text, it is unclear what these samples refer to. The methods section "building aggregate networks" seems to lack many details. For example, it is mentioned that the data for each consensus cell type is randomly split into samples of 20 cells each, and that the corresponding coexpression network is built based on "aggregate read counts". Does this mean that the 20 cell splits were added up and one coexpression network was built from the added samples? If so, please clarify this in the text.

Based on the above notes, the exact process of building these networks is unclear to me, however it seems that these networks are built to reflect the similarity of gene-gene expression patterns among cell types for each of the species.

We apologize for this confusion. The pseudobulk samples used to calculate the expressolog scores just average all cells within a consensus cell type. Since this approach leaves only one pseudobulk sample per cell type, we cannot use this to calculate cell type-specific coexpression networks schematically illustrated in Fig. 4A. The cell-type specific networks are, in spirit, simply co-expression between genes across cells within that cell-type. The one modification is that building a coexpression network directly from single cell data is challenging due to noise and sparsity inherent in snRNA-seq data. Therefore, we average counts from 20 nuclei together within a consensus cell type to yield a reasonable number of smoothed cell type-specific samples that can then be used to build a cell type-specific coexpression network. A key feature of this network is the lack of covariance due to shared changes in cell type (since all pseudobulk samples are from the same cell type). We have edited the Methods section for better clarity.

"The read counts are aggregated across 20 nuclei in each sample. The corresponding coexpression network is built by calculating the Spearman correlation between all pairs of highly variable genes across these pseudobulk samples, and then ranking the correlation coefficients for all gene-gene pairs, with NAs assigned the median rank. Aggregate single cell coexpression network for each primate is generated by averaging the rank-standardized networks from individual cell types. For example, the human single cell coexpression network is built by averaging the 57 cell type-specific networks (human|Astro_1, human|Oligo_9, human|Lamp5_2, human|Sst_20, human|L4 IT_10, human|L5 ET_1, etc.)."

It is not clear how paragraph 3 ("Are gene coexpression relationships...") relates to other parts of this section or to what the title implies. Paragraph 4 ("While the neocortex...") seems to have started abruptly.

We agree that the original text was not clear about the connection between paragraphs 3 and 4, and the rest of the section. We have edited the text in this section for better flow. An example of the excerpted text is added below.

"Currently, we need greater single-cell sequencing depth for better transcriptomic coverage, and multiple high-quality datasets that can be aggregated to build well-powered cell type-specific"

coexpression networks to link changes in gene expression to cell type-specific regulatory rewiring. Therefore, coexpression network analysis using large-scale aggregation of bulk expression data remains important for studying evolutionary differences driving species-specific transcriptional signatures. However, we need to evaluate the conservation of gene coexpression (i) across different levels of cell type granularity, and (ii) across divergent species before we can leverage the vast amounts of publicly available bulk RNA-seq data to pinpoint species-specific regulatory changes that could be associated with divergent expression patterning in the MTG.”

Paragraphs 3 (“Are gene coexpression relationships...”) and 4 (“While the neocortex...”) provide details on gene coexpression conservation across different cellular resolutions and across phylogenetically diverse species, respectively. Our results suggest that core transcriptional programs are shared across cell types and species, which allows us to select genes with human-specific expression and regulatory patterns through an integrative analysis of single cell and bulk transcriptomic data.

Regarding the coexpression conservation between different networks (Figure 4C), please elaborate on how these measurements can be interpreted.

Previously, we defined coexpression conservation as a measure of the number of connections (i.e. coexpression partners) shared by a gene between a pair of networks (Crow et al, *NAR* 2022). The coexpression conservation score is expressed as an AUROC, with a value of 1 indicating that the top-ranked coexpression partners are conserved across networks, 0.5 indicating random reordering of coexpression partners across networks, and 0 indicating that the top-ranked coexpression partners in one network are ranked at the bottom in the other network. High coexpression conservation of a gene suggests that its expression profile relative to other genes has not changed substantially across networks, which we interpret as a measure of gene functional conservation.

In order to distinguish coexpression (and in turn, conserved function) due to shared co-regulation from coexpression driven by cell type composition, we constructed gene coexpression networks using single cell data and compared them with coexpression networks built from bulk brain-specific and cross-tissue data. In Fig 4C, the high degree of consistency between single-nucleus and bulk networks reaffirms a model of multiscale coexpression in the brain (Harris et al, *Cell Systems* 2021), where genes that are highly variable across tissues also exhibit expression variation at finer resolution (across and within cell types in the brain), suggesting the presence of shared regulatory programs across cell types.

We have edited the text to add a brief explanation on interpreting the coexpression conservation measurements reported in Fig 4C.

“The high degree of consistency between single-nucleus and bulk networks highlights conserved co-regulatory relationships across tissues and cell types. This observation is consistent with a model of multiscale coexpression in the brain, which proposes that cell types may differ significantly in gene expression levels, but share a core co-regulatory network.”

Regarding the highly variable genes, how was the count 4500 selected for the coexpression analysis, and why is the count 3000 selected for the subclass label transfer?

We used the default settings for the label transfer function in Seurat v3, which selected 3000 highly variable genes. However, we would like to note that cell type signatures at the class and subclass levels are robust across species and brain regions, and can be identified with only a handful of high variance genes. Previously, we have reported that aggregate coexpression networks are better powered for genes expressed across all cell types compared to genes with cell type-specific expression (Crow et al, *NAR* 2022). For coexpression analysis using networks constructed from single cell, bulk brain and bulk RNA-seq datasets (reported in Fig 4C), we selected the top 4500 genes which are expressed in at least half the nuclei in the human MTG dataset.

Minor comment: Inconsistency in compound words - Figure 4A, 5A, "pseudobulk" and "pseudocell" vs. "pseudo-bulk" in the text.

Thank you for pointing out this inconsistency, we have edited the text and figures for uniformity.

Reviewer #2 (Remarks to the Author):

In this study, Suresh et al. analyze single nuclei from the middle temporal gyrus (MTG) of humans, chimps, gorillas, macaques, and marmosets to investigate evolutionarily conserved and divergent patterns of gene expression. The authors report 57 homologous cell types across the species, in which gene expression patterns are overwhelmingly conserved. Notably, a subset of genes exhibit human-specific patterns, evidenced by divergent signatures across one or more cell classes in single-nucleus data, and divergent regulatory relationships in cross-species bulk gene expression coexpression networks. Divergent genes were characterized as having higher loss-of-function tolerance compared to conserved genes, leading the authors to conclude that human-specific features in the MTG evolved under comparatively mild evolutionary constraints. Finally, the authors point to human-specific gene expression patterns in a disease-implicated gene (BBS5) as a potential source of phenotypic disparity between humans and animal models, thereby demonstrating the broader utility of their approach to predict translational success (or lack thereof) in animal models of disease.

The authors are studying an interesting topic using relatively new data modalities and have developed an impressive track record in this field. Figures are nicely prepared, but analytical threads are hard to follow.

We thank the reviewer for their thoughtful comments.

Major comments:

1) This study includes a large amount of data and many complex analyses. The authors need to provide more details in the methods and / or elsewhere to ensure that these analyses are clear and reproducible. In its current form, it is difficult to precisely understand all of the steps that were taken between raw data and the identification of 57 homologous cell types. Breaking down one paragraph (p. 22) as an example:

"Datasets from each species and modality (SSv4, Cv3) were analyzed independently. Briefly, each dataset was subset into 5 neighborhoods (CGE-derived and MGE-derived inhibitory neurons, IT type and deep excitatory neurons, and non-neurons) based on prior knowledge from human and mouse studies of cortical cell types (2, 23)." What prior knowledge and how was this done?

We thank the reviewer for pointing out that the original text did not provide adequate information on snRNA-seq quality control and processing. To address this issue, we have added a new subsection "Cell type label transfer" under the Methods section titled "Single nucleus RNA-sequencing processing and clustering" with further details on cell type annotation:

"Subclass annotations from the human primary motor cortex (M1) taxonomy (Bakken et al, Nature 2021) were used to annotate cell subclasses in the primate MTG by performing label transfer with Seurat v3 (Stuart et al, Cell 2019). Datasets were pre-processed with the standard LogNormalization method, followed by the selection of 3000 highly variable genes (or their orthologs for non-human primates) with the 'vst' method, and label transfer based on the first 30 PCs."

"Each neighborhood was annotated using the label transfer function from Seurat (59) with cell subclass labels from the recently published human primary motor cortex (M1) taxonomy (23)." Does the label transfer function have any parameters? What is the justification for using cell subclass labels from an entirely different brain region (M1) to annotate MTG and why is this not mentioned in the Results?

We apologize for not mentioning this in the Results or Methods sections. We have amended the text in both sections to provide further details on cell type annotation. An example of added text from the Results section is excerpted below.

"Since gene expression signatures at the cell subclass level are highly reproducible across species and brain regions (Bakken et al, Nature 2021), the primate MTG datasets were annotated by transferring subclass labels from the human primary motor cortex taxonomy (Bakken et al, Nature 2021)."

To answer the reviewer's specific questions regarding the label transfer function parameters: 3000 highly variable genes were identified with the 'vst' method on log-normalized data, and label transfer was performed using the first 30 PCs. Different IT types were difficult to predict using only label transfer since IT types from different cortical layers are transcriptomically very

similar. Additionally, due to the absence of L4 in M1, information from layer dissection was also used to annotate cell subclasses in human, chimpanzee and gorilla.

“Subclass label transfer was performed using 3000 highly variable genes or their orthologs for human and non-human primate datasets, respectively.” How exactly were these 3000 highly variable genes defined? Are they listed anywhere?

The 3000 highly variable genes were selected for each species using the 'vst' method in the FindVariableFeatures function of Seurat v3. Label transfer was performed without the exclusion list. For clustering and further analysis of each species' dataset, the highly variable genes were filtered to exclude sex and mitochondrial genes derived in Hodge et al (*Nature* 2019). The gene exclusion list is provided at:

https://github.com/AllenInstitute/Great_Ape_MTG/blob/master/exclusiongenes_mito_sex_tissue.txt, which is now included in the “Data and materials availability” section.

“Datasets underwent additional QC and passing nuclei from each dataset were normalized using SCTransform (60).” What kind of QC and how was this done? Any normalization parameters?

After low quality nuclei were filtered from single-nuclei RNA-seq datasets (SSv4 and Cv3), QC was performed at the neighborhood level. We have now provided further details on neighborhood-level QC in the Methods subsection titled “Filtering low-quality nuclei”.

“Next, QC was performed at the neighborhood level. Neighborhoods were split into over 100 metacells using Louvain clustering, and low-quality metacells with relatively low UMI or gene counts (glia and neurons with fewer than 500 and 1000 genes detected, respectively), predicted doublets (nuclei with doublet scores above 0.3), and/or low subclass label prediction metrics within the neighborhood (for example, excitatory labeled nuclei that clustered with majority inhibitory or non-neuronal nuclei) were removed from the dataset. Remaining high quality nuclei were normalized with SCTransform (version 1; (60)) using default parameters.”

“An integrated space was generated for each species by performing a canonical correlation analysis (CCA) across individuals and modalities. Each integrated space was clustered into hundreds of 'metacells', and metacells which passed quality control were merged with their nearest neighbors until merging criteria were met, resulting in the final clusters for each species (refer to Jorstad et al (26) for further details on RNA-seq processing, QC and annotation).” How was CCA performed? How was clustering performed? What type of cluster QC was performed? What were the merging criteria? Referring the reader to a dense bioRxiv manuscript for further details is not very helpful. It does not seem that the code underlying the steps described above has been made available in the GitHub repo or elsewhere, but please correct me if I am wrong.

We have now added a new subsection “snRNA-seq clustering” in the Methods to elaborate on dataset integration and clustering.

“Neighborhoods across individuals and modalities within a species were integrated together by identifying mutual nearest neighbor anchors and applying canonical correlation analysis (CCA) as implemented in Seurat v3. Smart-Seq v4 dataset (where available) was treated as an individual donor in the integration strategy. For example, deep excitatory neurons from human-Cv3 were split by individuals and integrated with the human-SSv4 deep excitatory neurons. The SelectIntegrationFeatures function was used to identify 3000 genes for integration, and datasets were integrated across the first 30 PCs. Sex and mitochondrial genes from the gene exclusion list were removed from the list of 3000 genes used for integration. The gene exclusion list used in this study was derived in Hodge et al (Nature 2019), and can be accessed from:

https://github.com/AllenInstitute/Great_Ape_MTG/blob/master/exclusiongenes_mito_sex_tissue.txt

The integrated space containing the remaining genes was then scaled and projected into 30 PCs, which were used for the clustering of each neighborhood. Each neighborhood was clustered using a previously described ‘shatter and merge’ approach (Bakken et al, Nature 2021). Louvain clustering was performed using the FindClusters algorithm from Seurat with variable resolution parameters until over 100 clusters, or ‘metacells’ were identified for each neighborhood. Metacells were merged with their nearest neighbor until each metacell contained more than 20 nuclei, and had a total of 8 genes (4 for glia) or more differentially expressed with every other metacell. Here, differentially expressed genes are defined as being expressed in more than half of nuclei in both metacells, and have a fold-change of 2 or more across the metacell pair, and have a proportion expressed differential of 0.3 or greater. The remaining clusters underwent further QC to exclude low-quality and outlier populations. These exclusion criteria were based on irregular groupings of metadata features that resided within a cluster.”

The primate MTG taxonomies used in this study required some manual curation and generated many intermediate files during QC, annotation and integration across individuals, modalities, species and cell type neighborhoods. Therefore, the code underlying the QC and clustering steps described above is not currently available on Github.

2) What is the relationship between AUROC scores, statistical significance, and multiple testing? If AUROC = 0.5 corresponds to chance, can one establish the ‘significance’ of arbitrary AUROC thresholds (e.g., 0.6 used for cross-species clustering) based on empirical null distributions of AUROC using permuted data, or otherwise? How is this impacted by multiple testing?

AUROC is an effect size which maps to the Mann-Whitney test statistic as follows:

$$AUROC = \frac{U}{n_0 * n_1}$$

where U is the Mann-Whitney U test statistic, n_0 is the number of positives (cells from the cell type of interest) and n_1 is the number of negatives (background cells). Therefore, AUROC for each pair of cross-species clusters can be converted to a p-value, followed by multiple hypothesis test correction to select significantly similar cell types across species. In our case,

since the numbers are very large, it is more useful to quote the effect size (i.e., any AUROC above 0.55 is nominally significant and 0.6 is definitely significant after an FDR correction). However, we would like to point out that our analysis is much more conservative since the AUROC of 0.6 (used in the cross-species clustering) is not the AUROC of the best hit (after evaluating all hits) but the best hit vs *only* the second best as an alternative. So, we are testing not just if there is a good hit (AUROCs for this case are close to 1), but if there is a uniquely good hit. We think the effect size is a good way of summarizing the results and setting an appropriate threshold which would generalize across studies better.

We agree with the reviewer that we can evaluate the significance of our AUROC thresholds based on empirical null distributions of AUROCs derived from permuted data. As an example, we permuted cell cluster labels in chimp data, and recomputed the cluster replicability AUROCs between human and chimp datasets. At an AUROC threshold of 0.6 in the stringent one_vs_best mode, we obtained no hits in the 100 permutations of chimp cell annotations, and only two hits at an AUROC threshold of 0.51. Cluster replicability scores in the one_vs_all mode were also very low (median of 0.5 in the permuted data vs 0.99 in the non-permuted data), and never rose above 0.79, as seen in the boxplot below.

Replicability of cell clusters between human and chimp MTG (MetaNeighborUS in one_vs_all mode)

We have added this explanation on the statistical significance of the AUROC threshold to the Methods section titled "Replicability of clusters".

3) All conclusions from the study are derived from the initial identification of homologous cell types, and therefore showing the existence of cell type replicability is paramount. While the authors take steps to validate cell types (leave-one-out cross-validation, MetaNeighbor), the study could benefit from either: i) further demonstrating the presence of homologous cell types in independent datasets from the same brain region that were not used in the study, or ii) histological validation of species-specific gene coexpression relationships.

We appreciate the reviewer's comment. To assess homologies between clusters from taxonomies across different studies, we compared the distance between cell type clusters in the original human MTG taxonomy derived from SSV4 nuclei (Hodge et al, *Nature* 2019) and

clusters from the human MTG taxonomy reported in this study. We computed the Euclidean distance between cluster centroids in the reduced dimensional space using 30-50 principal components from a PC analysis. Heatmap of log-transformed Euclidean distance between cluster centroids of human MTG cell types annotated in this study (columns; cell types labeled by class, neighborhood, and subclass) and cell types from a previously published human MTG taxonomy (rows; Hodge et al, *Nature* 2019) are shown below, with smaller values indicate greater transcriptomic similarity between taxonomies.

From the heatmap, we observe a good correspondence between the cell type clusters identified in the human MTG in this study and similar cell types identified in an independent study also sampling the human MTG, highlighting the replicability of our cell types across studies and giving us a higher confidence in our cell type annotation. We hope that our new higher resolution MTG taxonomy provides the community with a powerful resource to investigate conserved and divergent gene programs across primate brain evolution.

We agree that histological measurements of gene expression and coexpression patterns across species would help validate the uniquely human gene expression patterns observed in our datasets. With the rapid development of spatial transcriptomics techniques, we believe that our approach can provide a suitable gene panel to explore the divergence of spatial expression profiles across species.

4) Although the motivation for their analysis is clear, the most obvious evidence for evolutionary change may come from the things that are hardest to compare, i.e., putative cell types that are

present in one species but not others. Is there anything to be learned by studying the transcriptional phenotypes of non-homologous cell types?

We appreciate the reviewer's comment that evolutionary change may be driven by species-specific cell types. In our current study, we focused on detecting highly replicable cell types across species, so we did not identify any species-specific cell types. However, we did find 11 cell types each shared across four and two primates, and 7 cell types shared across three primates (shown in Figure S1A and C). Markers of non-homologous cell types in human (i.e. absent from 57 consensus cell types) were enriched for synapse organization and signaling, consistent with the prevalent idea that non-neuronal cell types are highly diverged in humans (Jorstad et al 2022, <https://doi.org/10.1101/2022.09.19.508480>; Pembroke et al, *Genome Biol.* 2021). Although we are not currently powered to identify cell types present in one species but not others, we hope our datasets provide a useful resource to uncover species-specific cell types when aligned against higher resolution spatio-transcriptomic atlases that may become available in the near future. We now summarize the main points in the Discussion section.

Reviewer #3 (Remarks to the Author):

In the manuscript "Comparative single cell transcriptomic analysis of primate brains highlights human-specific regulatory evolution", Suresh and colleagues present an analysis of single-cell RNA sequencing (scRNAseq) data from the cerebral cortex of five primate species. The aim of the analysis is to identify genes with human-specific expression or co-expression profiles at the cell type level.

This manuscript reads as an extension of Jorstad et al (reference 26), a bioRxiv preprint that describes in detail the five new primate scRNAseq datasets including an integrated cell type taxonomy (the authors of this paper contributed to Jorstad et al).

Complementing Jorstad et al, this manuscript focuses on the evolution of gene coexpression. The authors identify 57 homologous cell types in primates. They then build coexpression networks from bulk RNA sequencing data and analyze network connectivity of the human-specific genes in the 57 homologous cell types. They find evidence for relaxed selective constraints for a small set of genes with human-specific cell type expression.

This manuscript will be relevant for readers interested in the evolution of gene expression patterns, cell types, and human-specific traits.

The manuscript is well written. I appreciated the figure schematics illustrating the logic of every data analysis step; this really helps navigate the paper.

From a bioinformatic and statistical perspective, the analysis seems sound and is well documented in the methods sections (although parts of the analysis are outside my immediate

field of expertise). Integrating single-cell and bulk RNAseq data is an interesting approach, and applying this approach to other contexts may lead to biological insights. The paragraph in the Discussion outlining the limitations of the study is extremely valuable.

My major point of concern is the interpretability and significance of the gene coexpression network analysis (figures 4 and 5).

We are grateful to the reviewer for their insightful and encouraging remarks, and agree that it is important to clarify the results of the gene coexpression network analysis.

Figure 4

Figure 4D reports that coexpression scores of 1:1 orthologs are conserved in metazoa. This is surprising and puzzling, because it implies that animals with a wide variety and divergence of cell types still share a large number of gene expression modules. I wonder whether the extent of conservation is overestimated because of the way the analysis is set up:

1. Selection of marker genes

What marker genes were included in the 1:1 orthologs set used for figure 4? How are they expressed across the primate cell type dataset? If the majority of the markers distinguish neurons vs non-neuronal cells, for example, then the results in figure 4 would be less surprising (enrichment of pan-neuronal specific coexpression sets). How sensitive are these results to the set of marker genes - in other words, do these results change significantly if the criteria for marker gene selection change? Does this set show significant enrichment for genes in specific functional categories? Anything specific to point out about the outliers in fig 4B? (For example: In what cell types are these expressed? Do they belong to specific functional categories?)

In our current analysis (reported in Fig 4D), we use a set of 1681 human markers (comprising 582 class, 929 subclass and 170 consensus cell type markers; list of marker genes provided in Table S4) to assess the extent of coexpression conservation with across primates (using single cell coexpression networks) and across metazoa (using bulk coexpression networks). This set of 1681 human marker genes are significantly enriched in molecular pathways related to synaptic signaling, synapse organization, and neuron development. As expected, the markers identified using human MTG atlas are also highly expressed across non-human primates, with broadly expressed markers exhibiting higher expression levels compared to cell type-specific markers.

The reviewer raises an excellent point regarding the effect of different marker gene selection criteria on coexpression conservation of marker genes in bulk networks. To assess this, we select a total of 200 markers at the level of cell class, subclass and consensus cell types. We observe that the cell type-specificity of marker genes has a small effect on the variation of coexpression conservation with species divergence time (marker genes in all cases are correlated with phylogenetic age with $P < 2.2e-16$). Increasing the size of the marker gene set (selecting 1000 vs 200 markers at each cell type granularity) only has a slight decrease in the correlation of gene coexpression conservation with species divergence time, confirming the robustness of coexpression conservation of marker genes to different selection criteria. We now include a Supplementary Figure (Fig. S2) to highlight this result, and reference this figure in the text.

Defining the outliers in Fig 4B as human genes with expressolog scores below 0.5 in at least one non-human primate, we find 2,582 outlier genes in a total of 14,131 genes (of which > 60% exhibit diverged expression profiles only between human and marmoset). Similar to the observation by Jorstad et al (<https://doi.org/10.1101/2022.09.19.508480>), we also find that the outlier genes are significantly associated with four major pathways: ribosomal processing, extracellular matrix, axon structure, and the synapse. Genes associated with ribosomal processing were specific to interneurons in humans, while genes related to the other three pathways were specific to different non-neuronal cell types.

2. Tissue heterogeneity in co-expression networks

A critical issue may lie in the fact that the coexpression networks are built from bulk RNAseq data from a variety of tissues, not just brain. There are many examples of genes defining neuronal identity (example: transcription factors, extracellular matrix) that are co-expressed with

completely different genes in different tissues, for example brain and muscle. To my understanding (but I might be wrong), those genes would get a low coexpression conservation score in this analysis, because the coexpression networks are built from a heterogeneous collection of tissues. Would it be possible to repeat the analysis using networks built on bulk brain data only?

The reviewer has raised an important question regarding the influence of tissue heterogeneity in quantifying gene coexpression conservation, and we have done two follow-up experiments to address this issue. We have previously reported that the robustness of meta-analytic coexpression networks increases with the number of experiments (Crow et al, *NAR* 2022). In that study, we constructed a reference and ten bootstrapped coexpression networks by aggregating individual experiments for each species, and evaluated the robustness of the reference network by measuring the ability of ranked edges in the bootstrapped networks to predict the top 1% of edges in the reference network. We observed that we require reference networks built from 20 or more experiments to achieve a performance of 0.99 (AUROC) or higher in this task.

In our first experiment, we built tissue-specific aggregate coexpression networks using 20 or more experiments (bulk RNA-seq datasets) sourced from the Gemma database (Zoubarev et al, *Bioinformatics* 2012). Due to the lack of availability of multiple, independent brain-specific datasets across diverse species, we restrict our analysis to brain-specific and blood-specific networks in human and mouse only for evaluation purposes. For each species, we built three aggregate coexpression networks using datasets from brain, blood, and other tissues, and a fourth network using all tissue-specific datasets. Overall, we observed high coexpression conservation of genes across different tissues within species (mean AUROC of 0.98), and across species (mean AUROC of 0.93 between human and mouse datasets). Additionally, we also observed a significant drop in coexpression conservation (mean AUROC ~ 0.5; figure not shown here) when we built brain-specific networks for species with five or fewer experiments, suggesting that the lack of power in tissue-specific datasets has a much greater influence on gene coexpression conservation than tissue heterogeneity inherent in bulk data. Thus, using variable tissue to capture more species at high power seems a useful strategy; our expectations based on previous work are what motivated these experimental design considerations.

Coexpression conservation across human and mouse aggregated tissue-specific datasets from GEMMA

Number of datasets aggregated for each tissue and species

	human	mouse
brain	26	153
blood	24	22
others	137	235
all	187	410

We agree with the reviewer that genes that are coexpressed with different gene sets in different tissues might exhibit lower coexpression conservation scores. However, our previous experiment comparing coexpression conservation across human brain and blood-specific networks suggests that genes predominantly have conserved coexpression neighborhoods across tissues (although their expression levels will vary across tissues). To further quantify the variability in gene coexpression conservation across different tissues, we built tissue-specific coexpression networks using the gene expression data (TPMs) across 30 tissues from the GTEx database (v8; Lonsdale et al, *Nat. Genet.* 2013). We observed that genes typically retained their coexpression partners across tissues (mean coexpression conservation score of 0.87 across all tissue pairs). As in the experiment we outlined above, these results indicate that variation in coexpression conservation has little dependency on tissue heterogeneity in bulk data. We now include a Supplementary Figure (Fig. S3) to highlight this result, and reference this figure in the text.

Of course, tissue-specific co-expression may also exist and would not be validated by our approach, but we think this broader-based evaluation works well for capturing initial differences between species.

Figure 5:

As the authors point out, the analysis underlying identification of the 139 genes with human-specific regulatory divergence is extremely conservative, and may underestimate the number of genes with human-specific divergence. This analysis was obviously human-centric. What would we see with, say, a gorilla-centric, or a marmoset-centric analysis? If analyses centered on other primate species were to yield a similar number of diverging genes, that would suggest that humans are not that special after all.

The reviewer raises a valid criticism that genes diverged in non-human primates might be comparable in number to humans, which could indicate a lack of selection in driving human brain evolution. To address this, we applied the same method to filter genes with expression and coexpression divergence specific to other non-human primates. Since we could only find 4 bulk RNA-seq datasets each to build aggregate coexpression networks for gorilla and marmoset, we restricted our analysis to selecting chimp-specific and rhesus macaque-specific diverged genes. Similar to humans, a large number of genes (3833 genes in chimps and 5552 genes in rhesus macaques) show diverged expression in one or more cell classes, but only 32 genes in chimps and 14 genes in rhesus macaques also show species-specific coexpression divergence in bulk networks. We also found only one chimp-specific divergent gene that is also divergent in

humans. Note that the higher number of genes with cell class-specific expression divergence in rhesus macaque could be due to differences in sequencing technology, species divergence times, or great ape-specific expression patterning. Although rhesus macaque, chimp and human coexpression networks are differently powered (aggregate of 116, 27 and 90 experiments respectively), we still find more human- and chimp-specific genes than macaque-specific genes, suggesting that the power of bulk coexpression networks does not drive species-specific differential coexpression. The chimp-centric and macaque-centric analysis also suggests greater transcriptional divergence in humans which could drive phenotypic novelty.

Next, we modified our filtering criteria to select a set of genes that could be diverged in one or more primates. Specifically, we repeated the analysis for each of the three species (human, chimp, rhesus macaque) after excluding the bulk coexpression conservation scores of the other two primates. With this approach, we found 142, 47, and 11 genes with differential expression and coexpression patterns in humans, chimps, and rhesus macaques, respectively. However, we only found six genes diverged in both human and chimp, one gene diverged in both human and macaque, and no genes diverged in all three primates. Together, our experiments tentatively suggest that a greater proportion of genes are under rapid evolution in the human lineage compared to non-human primates.

Decision Letter, first revision:

7th June 2023

Dear Jesse,

Your manuscript entitled "Comparative single cell transcriptomic analysis of primate brains highlights human-specific regulatory evolution" has now been seen by three reviewers, whose comments are attached. The reviewers believe the manuscript is improved but Reviewer #2 still raises a number of concerns which will need to be addressed before we can offer publication in Nature Ecology & Evolution. We will therefore need to see your responses to the criticisms raised and to some editorial concerns, along with a revised manuscript, before we can reach a final decision regarding publication.

I should stress that code deposition is a required condition for publication and that you must share your code and detailed information about manual curation so that the study can be reproduced.

We therefore invite you to revise your manuscript taking into account all reviewer and editor comments. Please highlight all changes in the manuscript text file in Microsoft Word format.

* If you have not done so already please begin to revise your manuscript so that it conforms to our Article format instructions at <http://www.nature.com/natecolevol/info/final-submission>. Refer also to any guidelines provided in this letter.

[REDACTED]

27Note: This URL links to your confidential home page and associated information about manuscripts you may have submitted, or that you are reviewing for us. If you wish to forward this email to co-authors, please delete the link to your homepage.

Nature Ecology & Evolution is committed to improving transparency in authorship. As part of our efforts in this direction, we are now requesting that all authors identified as 'corresponding author' on published papers create and link their Open Researcher and Contributor Identifier (ORCID) with their account on the Manuscript Tracking System (MTS), prior to acceptance. ORCID helps the scientific community achieve unambiguous attribution of all scholarly contributions. You can create and link your ORCID from the home page of the MTS by clicking on 'Modify my Springer Nature account'. For more information please visit www.springernature.com/orcid.

[REDACTED]

Reviewers' comments:

Reviewer #1 (Remarks to the Author):

I appreciate the authors' efforts in addressing all the comments with thoroughness and clarity. I have no further comments.

Reviewer #2 (Remarks to the Author):

I appreciate the authors' efforts to respond to comments from the initial review. A consistent theme among the initial comments was a lack of clarity around methods and analyses. Given that this study is entirely *in silico*, if analyses are unclear or results uninterpretable, there is not much left for the reader to go on. Although the authors have made a good-faith effort to address these concerns, this effort still feels incomplete. For example, even with the new methods section on snRNA-seq clustering, which is foundational for the entire study, the authors note that manual curation of MTG taxonomies and production of intermediate files during QC precluded sharing of the code used to perform these steps. I understand that these analyses can be messy, but this is tantamount to acknowledging that the foundational analyses of this study cannot be feasibly reproduced.

28I also continue to find it strange that the authors are using cell type definitions from human motor cortex to annotate cell types in the MTG, which seems to have been done for technical and not biological reasons. Motor cortex is functionally and cytoarchitecturally distinct from association cortex. I worry that anchoring primate MTG cell type annotations in human motor cortex space may somehow bias the authors' analyses and interpretations.

One way to short-circuit critiques such as these is through experimental validation. However, no predictions made by the authors' analyses have been experimentally validated. In the absence of novel experimental validation, one might use previously reported species differences as positive controls. However, efforts to pursue this path through the online resource were unsuccessful since most comparative analyses produced error messages.

Reviewer #3 (Remarks to the Author):

The authors revised the manuscript thoroughly and addressed all comments and concerns with edits to the text and additional analyses.

In response to my comments on figures 4 and 5, the authors added additional supplementary figures, that demonstrate that the analysis of marker gene coexpression across species is not sensitive to number of genes considered, nor to tissue heterogeneity.

Something that struck me in the authors' response is the fact that the number of genes used for the analysis in figure 4D is quite small (1681 genes). I was not expecting that to be the case. Indeed, the description of this analysis in the methods section is rather minimal. Please provide more details in the methods section, and indicate in the figure legends how many genes were used for the analyses shown in each panel. The title of figure 4 states "Coexpression conservation highlights shared gene regulatory landscape across metazoa", and without further details, it lends itself to overgeneralizations.

*****END*****

Author Rebuttal, first revision:

Reviewer #1 (Remarks to the Author):

I appreciate the authors' efforts in addressing all the comments with thoroughness and clarity. I have no further comments.

We greatly appreciate the insightful comments provided by the reviewer in the major review of our manuscript, and especially the suggestion to clarify the utility of gene coexpression network analysis in uncovering species-specific transcriptional patterns.

Reviewer #2 (Remarks to the Author):

I appreciate the authors' efforts to respond to comments from the initial review. A consistent theme among the initial comments was a lack of clarity around methods and analyses. Given that this study is entirely in silico, if analyses are unclear or results uninterpretable, there is not much left for the reader to go on. Although the authors have made a good-faith effort to address these concerns, this effort still feels incomplete. For example, even with the new methods section on snRNA-seq clustering, which is foundational for the entire study, the authors note that manual curation of MTG taxonomies and production of intermediate files during QC precluded sharing of the code used to perform these steps. I understand that these analyses can be messy, but this is tantamount to acknowledging that the foundational analyses of this study cannot be feasibly reproduced.

I also continue to find it strange that the authors are using cell type definitions from human motor cortex to annotate cell types in the MTG, which seems to have been done for technical and not biological reasons. Motor cortex is functionally and cytoarchitecturally distinct from association cortex. I worry that anchoring primate MTG cell type annotations in human motor cortex space may somehow bias the authors' analyses and interpretations.

We thank the reviewer for their comments on the manuscript. The human primary motor cortex (M1) taxonomy (described in Bakken et al, *Nature* 2021) was used to annotate cell types in the MTG at the subclass level through label transfer, since cell subclass annotations are consistent across species and cortical regions, as shown in a recent preprint (Jorstad et al. 2022, <https://doi.org/10.1101/2022.11.06.515349>).

To demonstrate the robustness of our results to different cell subclass annotation strategies, we selected marker gene sets for cell types in the human MTG taxonomy reported in Hodge et al (*Nature* 2019), and used these markers to annotate cell types at the subclass level for human and non-human primate MTG datasets described in Jorstad et al (*bioRxiv* 2022, <https://doi.org/10.1101/2022.09.19.508480>). As mentioned before, cell types in the human MTG are largely consistent across datasets (as seen in the heatmap of log-transformed Euclidean distance between cluster centroids of human MTG cell types annotated in Jorstad et al (columns; cell types labeled by class, neighborhood, and subclass) and cell types from a previously published human MTG taxonomy (rows; Hodge et al, *Nature* 2019)). We used this

mapping to assign subclass labels to cell type clusters reported in Hodge et al (Table S2 in that paper).

Note that we only identified 22 of 24 subclasses in the Hodge et al human MTG dataset since the smaller size of the dataset precluded us from uniquely identifying certain vascular and excitatory neuron IT cell types. We then used the MetaMarkers package (Fischer and Gillis, *iScience*, 2021) to select the top 100 marker genes for each of the 22 cell subclasses, which we first used to annotate the human MTG dataset reported in Jorstad et al.

Cell subclass annotations for cell type clusters reported in Hodge et al, and their corresponding marker gene sets can be accessed from: https://labshare.cshl.edu/shares/gillislabs/resource/Primate_MTG_coexp/Hodge_MTG_subclass_anno_marker_list.xlsx

We find that the subclass labels predicted using the human MTG taxonomy from Hodge et al are consistent with the subclass annotations predicted by label transfer with human M1 taxonomy used by Jorstad et al. Sankey diagram in panel A in the figure below shows the replicability of subclass labels annotated using MTG and M1 taxonomies (squares indicate the subclass annotations in each dataset, and gray lines indicate cell types with a high degree of transcriptional similarity across datasets. i.e., MetaNeighbor US one_vs_all AUROC > 0.9). We annotated the non-human primate MTG datasets in a similar manner, and found that the subclass annotations generated using the MTG (Hodge et al) and M1 taxonomies (reported in Jorstad et al) were mostly concordant (overall classification accuracy = 0.9, and adjusted Rand Index = 0.79). We also observed that cell types at the subclass level were almost perfectly

replicable across primates, irrespective of the annotation strategy, as seen in panel B (heatmap shows the 'one_vs_best' MetaNeighbor scores for 22 cell subclasses across primates, with cell types labeled by species and subclass. Each column shows the performance of a single training group across the five test datasets. Cell subclass replicability scores (AUROCs) are computed between the two closest neighbors in each test dataset, where the closer neighbor will have the higher score (shown in red), and all others are shown in gray (NA)).

Next, we re-calculated the expressolog scores for 14,131 genes at the subclass level (i.e. expression profile similarity of one-to-one orthologs across 22 cell types for each pair of primates), and compared them with the subclass-level expressolog scores calculated using subclass annotations provided by Jorstad et al. Again, average expressolog scores mirrored species divergence times, and were largely independent of subclass annotation strategy (evidenced by panels A and B). We also compared the expressolog scores between human and non-human primates for the 139 human-specific "Diverged" genes and the remaining "Conserved" genes (panel C). As expected, the genes diverged in humans had significantly lower expression profile similarity with their orthologs in both cases, highlighting the robustness of our results to differences in subclass labeling methods.

We have added a Methods section “Impact of different subclass annotation protocols on gene expression profile similarity” and Supplementary Fig. 6 to highlight the robustness of our analyses to different subclass annotation protocols.

One way to short-circuit critiques such as these is through experimental validation. However, no predictions made by the authors' analyses have been experimentally validated. In the absence of novel experimental validation, one might use previously reported species differences as positive controls. However, efforts to pursue this path through the online resource were unsuccessful since most comparative analyses produced error messages.

We apologize for this inconvenience. We have rectified this issue, and the online resource can now be accessed from https://gillislabs.shinyapps.io/Primate_MTG_coexp/.

Reviewer #3 (Remarks to the Author):

The authors revised the manuscript thoroughly and addressed all comments and concerns with edits to the text and additional analyses.

In response to my comments on figures 4 and 5, the authors added additional supplementary figures, that demonstrate that the analysis of marker gene coexpression across species is not sensitive to number of genes considered, nor to tissue heterogeneity.

We greatly appreciate the insightful comments provided by the reviewer in the major review of our manuscript, and especially the suggestion to incorporate appropriate control experiments to validate the robustness of our coexpression conservation analysis.

Something that struck me in the authors' response is the fact that the number of genes used for the analysis in figure 4D is quite small (1681 genes). I was not expecting that to be the case. Indeed, the description of this analysis in the methods section is rather minimal. Please provide more details in the methods section, and indicate in the figure legends how many genes were used for the analyses shown in each panel. The title of figure 4 states "Coexpression conservation highlights shared gene regulatory landscape across metazoa", and without further details, it lends itself to overgeneralizations.

Thank you for this suggestion. We have edited the figure caption for Fig. 4 to reflect the number of genes used for the analyses shown in each panel. We have also added a brief description of this analysis in the Methods section "*Calculating cross-species coexpression conservation*". We have also changed the title of Fig. 4 to "Using coexpression conservation to characterize ancient regulatory landscapes".

Decision Letter, second revision:

22nd June 2023

Dear Jesse,

Thank you for submitting your revised manuscript "Comparative single cell transcriptomic analysis of primate brains highlights human-specific regulatory evolution" (NATECOLEVOL-23020292B). Reviewer #2 wasn't able to look at the revision in detail but while they still think the study would benefit from more clarity around methods, analyses and figures and they don't agree with the decision to anchor MTG cell type definitions in M1, they don't want to stand in the way of publication. Therefore, we'll be happy in principle to publish your paper in Nature Ecology & Evolution, pending minor revisions to comply with our editorial and formatting guidelines.

Please email us a copy of the file in an editable format (Microsoft Word or LaTeX)-- we can not proceed with PDFs at this stage. We will then perform detailed checks on your paper and will send you a checklist detailing our editorial and formatting requirements in about a week. Please do not upload the final materials and make any revisions until you receive this additional information from us.

[REDACTED]

Our ref: NATECOLEVOL-23020292B

6th July 2023

Dear Dr. Gillis,

Thank you for your patience as we've prepared the guidelines for final submission of your Nature Ecology & Evolution manuscript, "Comparative single cell transcriptomic analysis of primate brains highlights human-specific regulatory evolution" (NATECOLEVOL-23020292B). Please carefully follow the step-by-step instructions provided in the attached file, and add a response in each row of the table to indicate the changes that you have made. Please also check and comment on any additional

35marked-up edits we have proposed within the text. Ensuring that each point is addressed will help to ensure that your revised manuscript can be swiftly handed over to our production team.

****We would like to start working on your revised paper, with all of the requested files and forms, as soon as possible (preferably within two weeks). Please get in contact with us immediately if you anticipate it taking more than two weeks to submit these revised files.****

In recognition of the time and expertise our reviewers provide to Nature Ecology & Evolution's editorial process, we would like to formally acknowledge their contribution to the external peer review of your manuscript entitled "Comparative single cell transcriptomic analysis of primate brains highlights human-specific regulatory evolution". For those reviewers who give their assent, we will be publishing their names alongside the published article.

Nature Ecology & Evolution offers a Transparent Peer Review option for new original research manuscripts submitted after December 1st, 2019. As part of this initiative, we encourage our authors to support increased transparency into the peer review process by agreeing to have the reviewer comments, author rebuttal letters, and editorial decision letters published as a Supplementary item. When you submit your final files please clearly state in your cover letter whether or not you would like to participate in this initiative. Please note that failure to state your preference will result in delays in accepting your manuscript for publication.

Cover suggestions

As you prepare your final files we encourage you to consider whether you have any images or illustrations that may be appropriate for use on the cover of Nature Ecology & Evolution.

Please submit your suggestions, clearly labeled, along with your final files. We'll be in touch if more

36information is needed.

Nature Ecology & Evolution has now transitioned to a unified Rights Collection system which will allow our Author Services team to quickly and easily collect the rights and permissions required to publish your work. Approximately 10 days after your paper is formally accepted, you will receive an email in providing you with a link to complete the grant of rights. If your paper is eligible for Open Access, our Author Services team will also be in touch regarding any additional information that may be required to arrange payment for your article.

Please note that *Nature Ecology & Evolution* is a Transformative Journal (TJ). Authors may publish their research with us through the traditional subscription access route or make their paper immediately open access through payment of an article-processing charge (APC). Authors will not be required to make a final decision about access to their article until it has been accepted. [Find out more about Transformative Journals](https://www.springernature.com/gp/open-research/transformative-journals)

Authors may need to take specific actions to achieve [compliance with funder and institutional open access mandates](https://www.springernature.com/gp/open-research/funding/policy-compliance-faqs). If your research is supported by a funder that requires immediate open access (e.g. according to [Plan S principles](https://www.springernature.com/gp/open-research/plan-s-compliance)) then you should select the gold OA route, and we will direct you to the compliant route where possible. For authors selecting the subscription publication route, the journal's standard licensing terms will need to be accepted, including [the journal's standard licensing terms](https://www.nature.com/nature-portfolio/editorial-policies/self-archiving-and-license-to-publish). Those licensing terms will supersede any other terms that the author or any third party may assert apply to any version of the manuscript.

[REDACTED]

Best regards,

[REDACTED]

37Reviewer #2:
None

Final Decision Letter:

2nd August 2023

Dear Jesse,

We are pleased to inform you that your Article entitled "Comparative single cell transcriptomic analysis of primate brains highlights human-specific regulatory evolution", has now been accepted for publication in Nature Ecology & Evolution.

Over the next few weeks, your paper will be copyedited to ensure that it conforms to Nature Ecology and Evolution style. Once your paper is typeset, you will receive an email with a link to choose the appropriate publishing options for your paper and our Author Services team will be in touch regarding any additional information that may be required

Due to the importance of these deadlines, we ask you please us know now whether you will be difficult to contact over the next month. If this is the case, we ask you provide us with the contact information (email, phone and fax) of someone who will be able to check the proofs on your behalf, and who will be available to address any last-minute problems . Once your paper has been scheduled for online publication, the Nature press office will be in touch to confirm the details.

Acceptance of your manuscript is conditional on all authors' agreement with our publication policies (see www.nature.com/authors/policies/index.html). In particular your manuscript must not be published elsewhere and there must be no announcement of the work to any media outlet until the publication date (the day on which it is uploaded onto our web site).

Please note that *Nature Ecology & Evolution* is a Transformative Journal (TJ). Authors may publish their research with us through the traditional subscription access route or make their paper immediately open access through payment of an article-processing charge (APC). Authors will not be required to make a final decision about access to their article until it has been accepted. [Find out more about Transformative Journals](https://www.springernature.com/gp/open-research/transformative-journals)

38Authors may need to take specific actions to achieve [compliance](https://www.springernature.com/gp/open-research/funding/policy-compliance-faqs) with funder and institutional open access mandates. If your research is supported by a funder that requires immediate open access (e.g. according to [Plan S principles](https://www.springernature.com/gp/open-research/plan-s-compliance)) then you should select the gold OA route, and we will direct you to the compliant route where possible. For authors selecting the subscription publication route, the journal's standard licensing terms will need to be accepted, including <https://www.nature.com/nature-portfolio/editorial-policies/self-archiving-and-license-to-publish>. Those licensing terms will supersede any other terms that the author or any third party may assert apply to any version of the manuscript.

We welcome the submission of potential cover material (including a short caption of around 40 words) related to your manuscript; suggestions should be sent to Nature Ecology & Evolution as electronic files (the image should be 300 dpi at 210 x 297 mm in either TIFF or JPEG format). Please note that such pictures should be selected more for their aesthetic appeal than for their scientific content, and that colour images work better than black and white or grayscale images. Please do not try to design a cover with the Nature Ecology & Evolution logo etc., and please do not submit composites of images related to your work. I am sure you will understand that we cannot make any promise as to whether any of your suggestions might be selected for the cover of the journal.

You can generate the link yourself when you receive your article DOI by entering it here: <http://authors.springernature.com/share>.

[REDACTED]

P.S. Click on the following link if you would like to recommend Nature Ecology & Evolution to your librarian <http://www.nature.com/subscriptions/recommend.html#forms>

** Visit the Springer Nature Editorial and Publishing website at http://editorial-jobs.springernature.com?utm_source=ejp_NEcoE_email&utm_medium=ejp_NEcoE_email&utm_campaign=ejp_NEcoE for more information about our career opportunities. If you have any questions please click [here](mailto:editorial.publishing.jobs@springernature.com).**